# Light-gated integrator for highlighting kinase activity in living cells

Wei Lin [1] ✉, Abhishek Phatarphekar[1], Yanghao Zhong[1,8], Longwei Liu [2], Hyung-Bae Kwon [3], William H. Gerwick[4], Yingxiao Wang[2], Sohum Mehta[1] & Jin Zhang [1,5,6,7] ✉

Protein kinases are key signaling nodes that regulate fundamental biological and disease processes. Illuminating kinase signaling from multiple angles can provide deeper insights into disease mechanisms and improve therapeutic targeting. While fluorescent biosensors are powerful tools for visualizing live-cell kinase activity dynamics in real time, new molecular tools are needed that enable recording of transient signaling activities for post hoc analysis and targeted manipulation. Here, we develop a light-gated kinase activity coupled transcriptional integrator (KINACT) that converts dynamic kinase signals into "permanent" fluorescent marks. KINACT enables robust monitoring of kinase activity across scales, accurately recording subcellular PKA activity, highlighting PKA activity distribution in 3D cultures, and identifying PKA activators and inhibitors in high-throughput screens. We further leverage the ability of KINACT to drive signaling effector expression to allow feedback manipulation of the balance of $G\alpha_s^{R201C}$-induced PKA and ERK activation and dissect the mechanisms of oncogenic G protein signaling.

Kinase signaling pathways stringently regulate cellular processes under normal physiological conditions, which are also central to tumorigenesis and cancer development when they are dysregulated. Thanks to the development of fluorescent biosensors for visualizing the activities of various target kinases, we are able to illuminate these biological signals and activities in situ in a real-time manner[1]. However, real-time biosensors face great challenges in several settings, particularly when it is desirable to remove the transiency of a real-time readout and convert a potentially transient cellular state into a "permanent" mark. For instance, when mapping the influence of complex tissue microenvironments on heterogeneous kinase activities, the ability to identify all cells across the entire tissue that experienced high kinase activity, even transiently, would be highly desirable, but is

unfeasible through live imaging of biosensors in real time. Furthermore, being able to selectively manipulate cell function based on the activity state of the cell also calls for the ability to convert a potentially transient cellular state into a "permanent" mark. The concept of a calcium memory dye, which converts calcium transients in single cells to their "permanent" fluorescence marking[2], was realized for the study of neuronal activity in the brain[3,4]. A similar design has been applied to the study of neuromodulator signaling[5]. These tools belong to a class of light-gated integrators where a specific sensing switch and a light-induced switch simultaneously control the recording of activity changes. Yet, such a light-gated integrator has been missing from the toolbox of available biosensors for illuminating kinase activity. Here, we present a light-gated kinase activity integrator design for

[1]Department of Pharmacology, University of California San Diego, La Jolla, CA, USA. [2]Alfred E. Mann Department of Biomedical Engineering, University of Southern California, Los Angeles, CA, USA. [3]Solomon H. Snyder Department of Neuroscience, Johns Hopkins University School of Medicine, Baltimore, MD, USA. [4]Center for Marine Biotechnology and Biomedicine, Scripps Institution of Oceanography and Skaggs School of Pharmacy and Pharmaceutical Sciences, University of California San Diego, La Jolla, CA, USA. [5]Moores Cancer Center, University of California San Diego, La Jolla, CA, USA. [6]Shu Chien - Gene Lay Department of Bioengineering, University of California San Diego, La Jolla, CA, USA. [7]Department of Chemistry and Biochemistry, University of California San Diego, La Jolla, CA, USA. [8]Present address: Division of Chemistry and Chemical Engineering, California Institute of Technology, Pasadena, CA, USA. ✉e-mail: wel328@health.ucsd.edu; jzhang32@health.ucsd.edu

converting kinase activity changes in live cells to a transcriptional change with high spatiotemporal resolution. This integrator displays robust performance in multiple research applications, including highlighting the distribution of basal PKA activities within 3D cultures, screening potential PKA inhibitors/activators from small molecule libraries, and inducing effectors for signaling network manipulation in living cells.

## Results

### Developing a light-gated integrator of protein kinase A activity

Light-gated, transcription-based activity integrators usually consist of three parts: a biochemical switch, an optical switch, and a transcriptional reporter. To construct a prototype kinase activity-coupled transcriptional integrator (KINACT) for memorizing protein kinase A (PKA) activity (A-KINACT), we used the PKA substrate (PKAsub; LRRATLVD) and the phospho-amino acid-binding domain forkhead-associated 1 (FHA1) of AKAR[6] as the biochemical switch to control reconstitution of split tobacco etch virus protease (TEVp). As the optical switch, we used a TEVp cleavage sequence (TEVseq) caged by a light-oxygen-voltage sensing (LOV) domain. Finally, we used a tetracycline-controlled transcriptional activator (tTA)/tetracycline operator (TetO) system as the transcriptional reporter. In the first component, PKAsub, the TEVp N-terminal fragment (TEVp-N; residues 1-118), the LOV domain-caged TEVseq, and the tTA are fused in tandem and targeted to the plasma membrane using a transmembrane (TM) helix derived from platelet-derived growth factor receptor beta (PDGFRβ). The second component comprises a cytosolically expressed fusion of the FHA1 domain and TEVp C-terminal fragment (TEVp-C; residues 119-220) (Fig. 1a). Induction of PKA activity in cells expressing A-KINACT will lead to PKAsub phosphorylation and binding of the FHA1 domain, resulting in close proximity of TEVp-N and TEVp-C, favoring TEVp reconstitution. Concurrent blue light irradiation of the LOV domain will expose the caged TEVseq, allowing cleavage by reconstituted TEVp and subsequent release of the tTA to drive reporter gene expression in the nucleus (Fig. 1a). To facilitate imaging and analysis, we used EGFP-tagged Histone 2B (H2B-EGFP) as the reporter.

To minimize potential leaky reporter expression due to TEVp-mediated cleavage in the absence of either kinase activity or light, we evaluated several configurations featuring different combinations of LOV domain variants and TEVp-TEVseq pairs that had been previously evolved and optimized in various integrators such as iTango[5], FLARE[4] and SPARK[7]. By comparing the H2B-EGFP+ fraction in HEK293T cells lacking expression of Component 2 (Construct 7), we found that TEVseq with Gly preceding the cleavage site (TEVseq-G, Constructs 1 and 4; Supplementary Fig. 1) showed little cleavage by endogenous proteases, whereas TEVseq with Met preceding the cut site (TEVseq-M, Constructs 3 and 5) exhibited poor bio-orthogonality and a weaker LOV-dependent response. We then transfected both Components 1 (Constructs 1–5) and 2 (Construct 7) into HEK293T cells and compared the fraction of H2B-EGFP+ cells in the absence (untreated/dark) or presence of PKA activation using Fsk and IBMX (F/I) plus blue light (treated/light). hLOV1-tethered TEVseq-G (Construct 1) achieved the best performance in terms of contrast between OFF signal in the untreated/dark state (2.7% H2B-EGFP+), which was comparable to the negative control TEVseq-P (Construct 2), and ON signal (28.1% H2B-EGFP+) in the treated/light state. In contrast, iLID-tethered TEVseq-G (Construct 3) exhibited both higher leaky expression (4.5% H2B-EGFP+) in the untreated/dark state and lower ON signal (18.5% H2B-EGFP+) in the treated/light state. Notably, we also found that an alternative configuration based on recruitment of intact TEVp (Constructs 6 and 8) showed very high basal signal (55.1% H2B-EGFP+) in the untreated/dark state. Therefore, we selected Construct 1 and Construct 7 as the main components of A-KINACT (Supplementary Fig. 1).

To facilitate expression of Components 1 and 2 at an appropriate ratio, as well as provide a convenient internal marker for quantification, we combined the two main components plus an mCherry marker into a single polycistronic cassette (Fig. 1b). Specifically, we placed mCherry after a P2A sequence to indicate the expression level of Component 1, with Component 2 driven by an IRES to promote reduced expression relative to Component 1, thus minimizing spontaneous binding. Indeed, HEK293T cells transfected with the polycistronic construct expressed Components 1 and 2 at roughly a 10:1 ratio (Supplementary Fig. 2). Meanwhile, we generated and characterized a HEK293T line stably integrating the TetO-H2B-EGFP reporter construct to avoid variability from transient transfection (Supplementary Fig. 3a, d). We then tested the performance of A-KINACT via epifluorescence microscopy under different conditions. Transiently expressing A-KINACT in TetO-H2B-EGFP cells produced low background fluorescence in the dark state (untreated, 0.7% H2B-EGFP+; treated, 0.9% H2B-EGFP+), with clear and varied signal increases−18.6 fold (no F/I, 4.5% H2B-EGFP+) and 65.0 fold (F/I, 11.4% H2B-EGFP+)−upon illumination with blue light (465 nm) for 20 min (10 s on/50 s off) (Supplementary Fig. 4a−c, also see Methods). However, when testing whether the reporting behavior of A-KINACT was dependent on phosphorylation of the PKA substrate by generating a non-phosphorylatable mutant (T/A) A-KINACT (Fig. 1b), we noticed that the mutant integrator produced a detectable increase in H2B-EGFP expression−8.4 fold (no F/I, 3.4% H2B-EGFP+) and 8.4 fold (F/I, 3.8% H2B-EGFP+) changes−under all illumination conditions versus the untreated/dark condition (Supplementary Fig. 4a, b, d). Given that H2B-EGFP expression did not increase upon F/I addition, we hypothesized that excessive A-KINACT expression in a fraction of cells, arising from transient transfection, may be driving spontaneous reconstitution of split TEVp and subsequent reporter expression. To test this possibility, we generated a TEV-only reporter system in which the PKAsub and FHA1 were omitted. Indeed, under blue light illumination, this TEV-only reporter induced a tenfold change in H2B-EGFP expression, similar to the T/A mutant A-KINACT (Supplementary Fig. 5).

Therefore, we established cell lines that stably express appropriate levels of each of the integrator variants (A-KINACT and T/A mutant A-KINACT) to avoid such overexpression-related artifacts in subsequent applications. In both the presence and absence of PKA stimulation, dual-stable A-KINACT cells exhibited similar low background (untreated, 2.0% H2B-EGFP+; treated, 2.5% H2B-EGFP+) in the dark, indicating tight control by light. Upon application of blue light, a large population (74.9%) of brightly H2B-EGFP+ cells were visible following F/I stimulation, representing a 98.0-fold change versus untreated/dark conditions (Fig. 1c–e). A small fraction (10.7%) of cells was also observed to express H2B-EGFP in the absence of F/I stimulation, representing a 7.8-fold change in H2B-EGFP expression versus untreated/dark conditions. To determine whether this EGFP+ cell population was still dependent on PKA activity, we pretreated cells with the PKA-specific inhibitor H89 prior to F/I and light application. We observed a smaller (5.8%) fraction of dim H2B-EGFP+ cells compared with the no-drug (light-only) condition (Fig. 1c–e), suggesting this EGFP+ cell population was at least partially dependent on basal PKA activity. In addition, A-KINACT (T/A) dual-stable cells presented extremely low background (untreated, 1.5% H2B-EGFP+; treated, 1.1% H2B-EGFP+) in the dark, and nonsignificant changes−0.7 fold (no F/I, 1.0% H2B-EGFP+), 0.2 fold (F/I, 0.4% H2B-EGFP+) and 0.7 fold (H89 + F/I, 1.2% H2B-EGFP+)−under all illumination conditions (Fig. 1c, d, f). Altogether, these results suggest that A-KINACT is an effective light-controlled tool for memorizing PKA activity changes.

### Characterizing the sensitivity and spatiotemporal resolution of A-KINACT in live cells

To characterize the sensitivity of A-KINACT, we performed dose-response experiments to evaluate the PKA activity enhancement

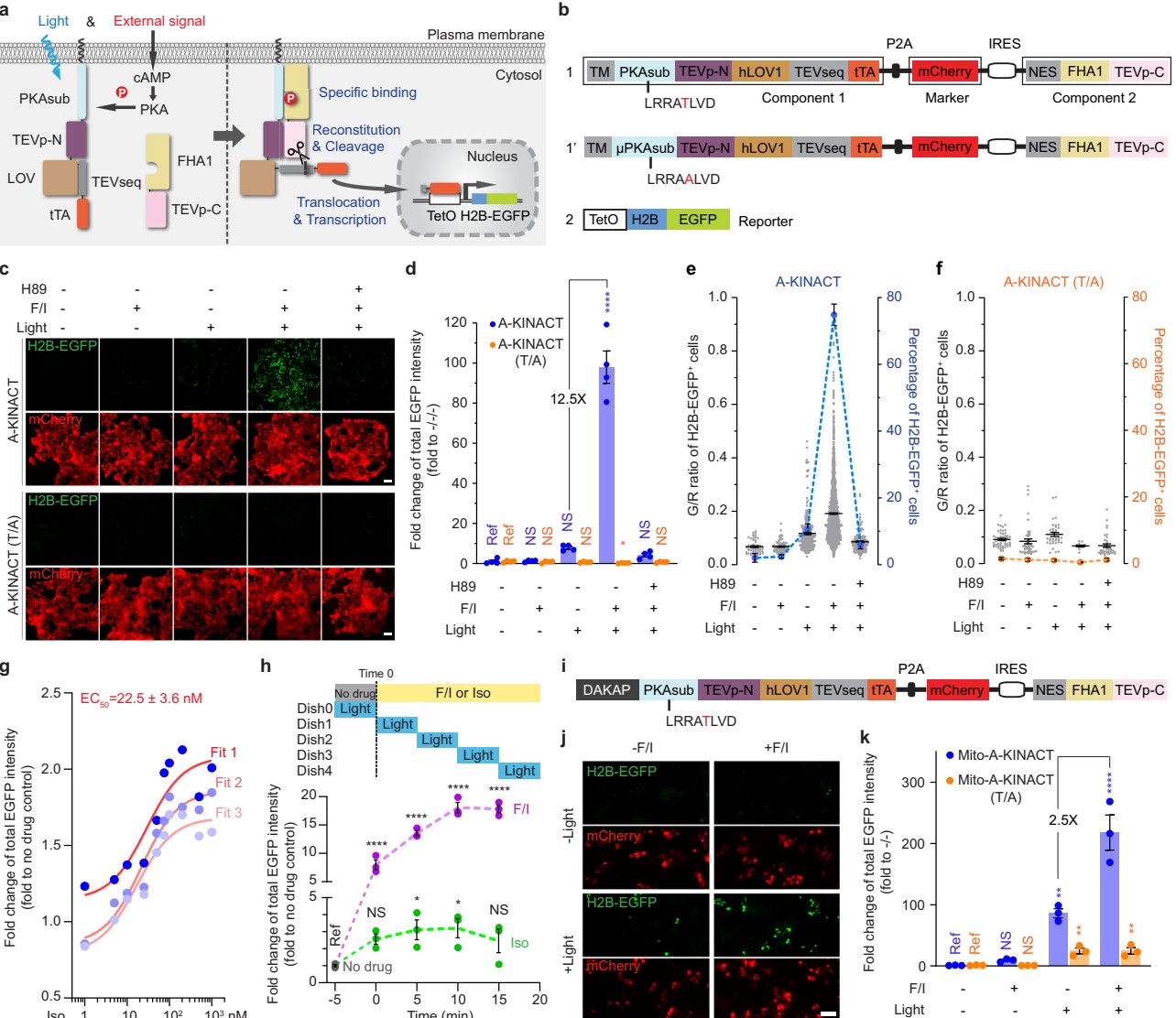

**Fig. 1 | KINACT for cumulative PKA activity recording in live cells. a** Schematic of PKA activity integrator and diagram of light- and kinase activity-induced gene expression. **b** The domain structures of A-KINACT (1), A-KINACT (T/A, 1') and reporter (2). **c** Snapshot imaging of cumulative PKA activity in A-KINACT and A-KINACT (T/A) dual-stable cells under five treatment conditions. **d**, Statistical quantification of total EGFP intensity under all conditions. *$P = 0.0388$ (T/A, +F/I/+light) and ****$P = 9.99 \times 10^{-16}$ (A-KINACT, +F/I/+light). **e** The fraction and mean EGFP/mCherry (G/R) intensity ratio of H2B-EGFP⁺ cells stably expressing A-KINACT. $n = 52$, $n = 73$, $n = 285$, $n = 1975$ and $n = 154$ cells. **f** The fraction and G/R ratio of H2B-EGFP⁺ cells stably expressing A-KINACT (T/A). $n = 52$, $n = 47$, $n = 37$, $n = 13$ and $n = 36$ cells. **g** The quantification of 3 independent Iso dose-response experiments in A-KINACT cells. The red lines represent 3 fitted logistic curves with an average fit value for the EC₅₀. **h** The quantification of light-gated experiments with full-time drug treatment (F/I, 50/100 μM or Iso, 100 nM) but 5 min time window of light

illumination. Data points correspond to start of illumination (min after drug addition). *$P = 0.0417$ (Iso, 5–10 min), *$P = 0.0333$ (Iso, 10–15 min), ****$P = 7.08 \times 10^{-5}$ (F/I, 0–5 min), ****$P = 1.33 \times 10^{-7}$ (F/I, 5–10 min), ****$P = 1.82 \times 10^{-13}$ (F/I, 10–15 min) and ****$P = 7.46 \times 10^{-10}$ (F/I, 15–20 min). **i** Domain structure of outer mitochondrial membrane-targeted A-KINACT (Mito-A-KINACT) system. DAKAP motif was tethered to the N-terminus of A-KINACT (1). **j** Snapshot imaging of cumulative PKA activity near mitochondria under four treatment conditions. **k** Statistical quantification of total EGFP intensity under all conditions. **$P = 0.0094$ (Mito-A-KINACT, -F/I/ + light), ****$P = 3.88 \times 10^{-6}$ (Mito-A-KINACT, +F/I/ + light), **$P = 0.0042$ (T/A, -F/I/ + light) and **$P = 0.0041$ (T/A, +F/I/ + light). For (**d**–**f**) data from 4 independent experiments. For (**g**, **h**) and (**k**) data from 3 independent experiments. For (**c**) and (**j**) scale bars, 10 μm. For (**d**–**h**) and (**k**) statistical analysis was performed using ordinary one-way ANOVA followed by Dunnett's multiple-comparisons test. NS, not significant. Data are mean ± s.e.m. Source data are provided as a Source Data file.

reported by A-KINACT in response to increasing concentrations of isoproterenol (Iso, β-adrenoreceptor agonist). We recorded PKA activity changes during a 20 min period after Iso addition and observed a graded increase in the H2B-EGFP signal corresponding to the applied Iso concentration (Fig. 1g). Using the resulting dose-response curve, we calculated an EC₅₀ value of 22.5 ± 3.6 nM, which falls well within the previously reported EC₅₀ range of 21–26 nM[8] for Iso stimulation of endogenous β2-adrenoreceptors. Our results suggest that A-KINACT is sensitive and accurate enough to report low-amplitude, physiological PKA signaling events.

In addition to sensitivity, we also assessed whether A-KINACT can accurately recapitulate the temporal dynamics of PKA activity changes. We therefore performed experiments in which A-KINACT HEK293T cells received drug treatment for a full 20 min duration but were illuminated with blue light for only a 5 min (10 s on/10 s off) time window starting at varying times after drug addition (Fig. 1h). As expected, cells treated with F/I showed a sustained increase in the H2B-EGFP signal, whereas Iso stimulation only induced a transient increase in H2B-EGFP fluorescence. Tracing the accumulation of fluorescence during each time window reflects the dynamic changes in PKA activity

induced under the different stimulation conditions, which are largely consistent with past results obtained using real-time PKA biosensors[9,10]. These data reveal that the two components of the KINACT system can reversibly combine and separate in response to changes in kinase activity, thus providing an accurate readout of kinase activity levels during the corresponding recording window.

Similar to previous integrators[3,4], our initial KINACT implementation tethers Component 1 to the plasma membrane (Supplementary Fig. 6a, c). Yet we also anticipated that the KINACT system can be applied to report kinase activity at other subcellular locations. We therefore targeted Component 1 of A-KINACT to the mitochondrial outer membrane using a peptide motif derived from DAKAP1 (Supplementary Fig. 2 and Supplementary Fig. 6b-c, see Methods) to sense and record PKA activities near the mitochondrial surface (Fig. 1i). Similar to the plasma membrane-targeted reporter, we observed an extremely low background signal in the dark and a significant (218-fold) induction of H2B-EGFP expression under concurrent F/I and light treatment (Fig. 1j, k and Supplementary Fig. 7). Compared with the plasma membrane-targeted version, mito-targeted A-KINACT reported a reduced PKA activity change from basal state to F/I stimulated state (2.5-fold versus 3.5-fold), which is consistent with previous results[9]. These data suggest that the KINACT system can be applied to monitor kinase activity within specific subcellular compartments.

## General applicability of the KINACT system

On the basis of traditional real-time biosensor toolbox design, we also expected KINACT to be broadly adaptable for recording various kinase activities. We first investigated whether KINACT could be generalized to other serine/threonine kinases by replacing the PKA substrate in A-KINACT with a protein kinase C (PKC)[11] substrate (Fig. 2a). The resulting PKC integrator (C-KINACT) successfully recorded PKC activity changes when expressed in HEK293T cells (Fig. 2b and Supplementary Fig. 2). We observed a 32-fold change in H2B-EGFP expression when cells were treated with phorbol 12,13-Dibutyrate (PDBu) and blue light for 20 min. Pretreating cells with the PKC inhibitor Gö6983 abolished the response to PDBu, indicating the specificity of C-KINACT. Conversely, cells expressing a T/A mutant C-KINACT showed no obvious response to PDBu treatment (Fig. 2c). To further investigate the general applicability of KINACT to record tyrosine kinase activity, we incorporated a Fyn kinase substrate and high-specificity SH2 domain[12] into Components 1 and 2, respectively, to generate a Fyn integrator (F-KINACT) (Fig. 2d and Supplementary Fig. 2). HeLa cells stably integrating TetO-H2B-EGFP were generated and characterized (Supplementary Fig. 3b, d). F-KINACT, but not a non-phosphorylatable Y/F mutant, produced obvious H2B-EGFP expression increases upon human EGF addition and blue light illumination in HeLa TetO-H2B-EGFP cells (Fig. 2e). Pretreating F-KINACT cells with PP1, a selective inhibitor of Fyn, also prevented EGF-induced H2B-EGFP expression (Fig. 2e). Quantification of total EGFP intensity revealed a 10.5-fold change compared with no drug and no light conditions (Fig. 2f). Together, both C-KINACT and F-KINACT highlight the potential for developing a broad KINACT toolkit for recording kinase activities.

## A-KINACT reveals intra-spheroid PKA activity distribution

The capability of KINACT to enable *post hoc* imaging and analysis of recorded kinase activities motivated us to peer into the distribution of PKA activity in 3D-cultured spheroids. Single cells stably expressing A-KINACT or A-KINACT (T/A) were cultured into spheroids with 5 or 6 cell layers on Matrigel, after which basal PKA activities were recorded during 20 min of blue light illumination. Under both epifluorescence and confocal microscopy, 3D-cultured HEK cells displayed obvious H2B-EGFP expression throughout the spheroid, whereas normal 2D cultured cells under the same illumination conditions showed very low basal PKA activity (Supplementary Fig. 8). These data suggest that cells

within a 3D tissue environment have upregulated PKA activity in the basal state. To visualize the distribution of this altered PKA activity, spheroids were scanned along the Z-axis via confocal microscopy to construct a 3D image with multiple layers (Fig. 3a). We observed high H2B-EGFP expression across multiple cell layers of the A-KINACT spheroids, with differential expression in individual cells (Fig. 3b and Supplementary Fig. 9a), in contrast to extremely weak H2B-EGFP expression within H89-treated A-KINACT spheroids or A-KINACT (T/A) spheroids (Supplementary Fig. 9b-c). We noticed a reduction in fluorescence intensity at greater imaging depths, which may be due to light scattering and absorption. Upon application of attenuation correction to correct the intensity of each channel, the distribution of H2B-EGFP+ cells appeared to be random in each layer without obvious differences between upper and lower layers (Supplementary Fig. 10). Application of A-KINACT for 3D imaging similarly revealed high basal PKA activity in MCF-7 spheroids (Supplementary Fig. 11). These data suggest that A-KINACT holds the potential to illuminate the spatial distribution of kinase activities inside complex, 3D tissue environments.

## High-throughput compound screening using A-KINACT

Given that KINACT is designed to integrate kinase activity into an amplified, transcriptional readout, we expected that A-KINACT would be capable of robustly detecting weak and transient modulators of PKA activity. Furthermore, in contrast to real-time recording, *post hoc* measurement is an important technological advantage of A-KINACT for screening compound libraries at various scales. As an initial proof of concept, we collected a small panel of well-characterized GPCR agonists known to activate, inhibit, or have no effect on PKA activity. Using epifluorescence microscopy, we found that HEK293T cells expressing A-KINACT responded strongly to several $G\alpha_s$-linked agonists, such as Iso, PGE1, and PGE2, reporting dose-dependent activity changes (Fig. 4a), which was validated using the real-time sensor AKAR4[13] (Supplementary Fig. 12a). We also detected the suppression of Fsk-stimulated PKA activity in response to the $G\alpha_i$-linked agonists lysophosphatidic acid (LPA) and Angiotensin II (Ang II) (Fig. 4b), which was validated using ExRai-AKAR2[9] (Supplementary Fig. 12b). Interestingly, we found that high doses of the $G\alpha_q$-linked agonist ET-1 (100 nM) led to transient increases in PKA activity (Supplementary Fig. 12c), which was successfully captured by A-KINACT (Fig. 4c). One possible reason is that PKA was stimulated by these $G\alpha_q$-linked agonists through parallel signaling such as calcium or PKC[14]. In addition, no obvious PKA activity was detected in response to treatment with Dopamine, GLP1, or Histamine, whose cognate receptors are known to be absent or underexpressed in HEK293T cells (https://www.ncbi.nlm.nih.gov/geo/query/acc.cgi?acc=GSM3734788)[15]. Thus, A-KINACT accurately reported the effects of different GPCR agonists on PKA activity.

Next, to test the feasibility of measuring KINACT responses using an automated multi-well plate reader and thus facilitate high-throughput compound library screening, we performed a pilot experiment by measuring the responses of A-KINACT to a small panel of activators or inhibitors and found that A-KINACT provides a sensitive readout regardless of the detection modality (Supplementary Fig. 13). We then screened a library containing 160 known kinase inhibitors, which were applied to A-KINACT-stable HEK293T cells alongside 5 μM Fsk, as well as blue light illumination for 20 min. Screening on a plate reader revealed 11 inhibitors that diminished Fsk-stimulated induction of PKA activity by >20% (Fig. 4d and Supplementary Data 1). These potential PKA inhibitors included three AKT inhibitors, two Chk1 inhibitors, one PKC inhibitor, one ATR inhibitor, one PKD inhibitor, one AXL family inhibitor, one ETK/BMX inhibitor, and one broad-spectrum kinase inhibitor. Since most of these compounds show some intrinsic fluorescence, we chose to use a pair of red-shifted, real-time PKA biosensors, GR-AKARev[16] and Booster-PKA[17],

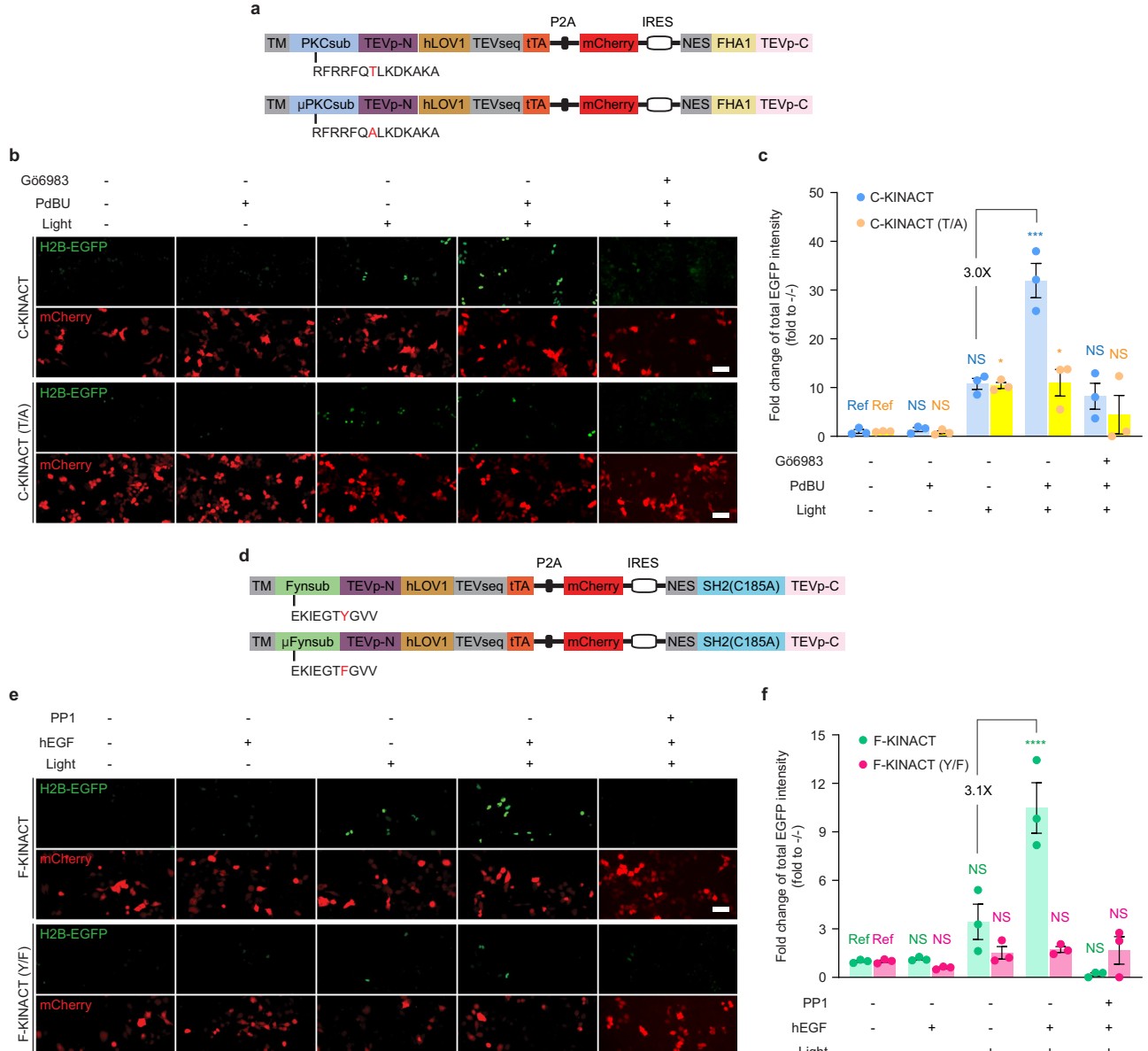

**Fig. 2 | General applicability of KINACT to different types of kinases. a** Domain structures of C-KINACT and C-KINACT (T/A). **b** Representative images showing C-KINACT and mutant C-KINACT (T/A) reporting PKC activity changes induced by PDBu (200 nM) and inhibited by Gö6983 (10 μM) in HEK293T cells. **c** Statistical quantification of total EGFP intensity under all conditions. ***P = 0.0003 (C-KINACT, +PdBU/+light), *P = 0.0233 (T/A, -PdBU/+light), *P = 0.0271 (T/A, +PdBU/+ light). **d** Domain structures of F-KINACT and F-KINACT (Y/F). **e** Representative images showing F-KINACT reporting Fyn activity changes induced by human EGF (hEGF, 100 ng ml⁻¹) and inhibited by PP1 (10 μM) in HeLa cells. **f** Statistical quantification of total EGFP intensity under all conditions. ****P = 2.39 × 10⁻⁵ (F-KINACT, +hEGF/+light). For (**c**) and (**f**) data from 3 independent experiments. For (**b**) and (**e**) scale bars, 10 μm. For (**c**) and (**f**) statistical analysis was performed using ordinary one-way ANOVA followed by Dunnett's multiple-comparisons test. NS not significant. Data are mean ± s.e.m. Source data are provided as a Source Data file.

to validate the inhibitory effects of these candidates. Of the 11 compounds, 10 showed inhibition of PKA activation by the maximum dose of F/I, largely in agreement with the A-KINACT data (Supplementary Fig. 14). One compound (1B3), which just cleared the cut-off in the A-KINACT results, showed a weak inhibitory effect in validation experiments but was not statistically significant. These data demonstrate the utility of A-KINACT in compound screens and further confirm the promiscuity of kinase inhibitors[18].

Aside from inhibitors, identifying novel PKA activators can potentially enhance the therapeutic arsenal against cardiovascular[19] and neurological diseases, while also helping screen out potential anticancer drug candidates that may produce adverse effects through unwanted PKA activation, thus creating a more effective therapeutic

pipeline[20]. Marine natural products are often heralded as a promising reservoir for drug discovery. We therefore screened a collection of 137 marine natural products using A-KINACT and identified 14 compounds that led to an at least 1.5-fold induction of PKA activity relative to the basal signal (Fig. 4e and Supplementary Data 2). Among these, we took particular note of four interesting compounds that suggest the modulation of PKA activity by certain key pathways. For instance, Psammaplin A, derived from a marine sponge, inhibits class I histone deacetylase (HDAC1)[21]; Kalkitoxin induces cellular hypoxia by inhibiting the electron transport chain (ETC) complex 1[22]; and Crossbyanol B and (7Z,9Z,12Z)-octadeca-7,9,12-trien-5-ynoic acid were reported as inhibitors of Cathepsin B[23]. As these identified PKA activators show relatively subtle effects, we verified these compounds using real-time

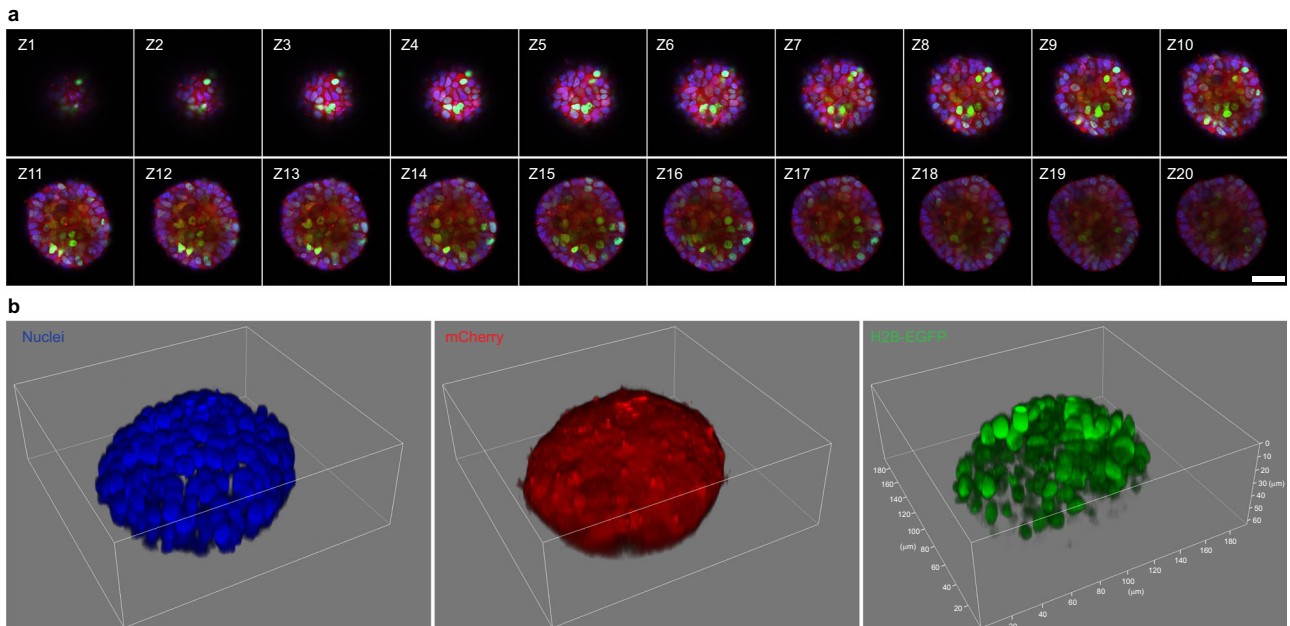

**Fig. 3 | 3D imaging of heterogeneous PKA activity in HEK293T spheroids.**
**a** Triple-color confocal imaging of an A-KINACT dual-stable spheroid in the Z-direction. 20 layers of merged images of Hoechst-stained nuclei (blue), mCherry (red) and H2B-EGFP (green). Scale bar, 50 μm. **b** 3D reconstruction of individual channels showing nuclear positions (left), uniform A-KINACT expression (middle) and distribution of PKA activity within the spheroid (right). $n = 3$ spheroids (see Supplementary Fig. 9a for Spheroids 2 and 3).

imaging in HEK293T cells expressing a sensitive PKA activity biosensor, AKAR3ev[24], and indeed observed definite bioactivities toward PKA within 20 min of drug addition (Supplementary Fig. 15). Taken together, our screening results confirm that the KINACT system supports high-throughput functional compound screening with high sensitivity and reliability.

### Applying KINACT to dissect oncogenic G protein signaling

The stimulatory G protein α subunit (Gα$_s$) regulates a variety of cell functions[25,26]. In recent years, activating mutations in the Gα$_s$-encoding gene GNAS have been detected in multiple cancer types, such as pancreatic and colorectal cancers[27–29]. The oncogenic Gα$_s$ R201C mutation leads to high PKA activity, which can be further stimulated via GPCR activation[30]. PKA is capable of triggering signaling cascades and regulating transcription of genes critical for cell proliferation. We recently discovered that Gα$_s$ can play an essential role in GPCR-mediated activation of the ERK pathway, which is known to play a critical role in regulating gene transcription and cell proliferation[31]. Given the complex crosstalk observed between the PKA and ERK signaling pathways[32], it is challenging to dissect how Gα$_s$^{R201C} reorganizes the PKA and ERK signaling networks to impact tumorigenesis, as well as the specific contributions of PKA versus ERK pathways. We therefore leveraged the innate ability of KINACT to translate cellular kinase activity into specific protein expression to engineer a KINACT effector that will exert negative-feedback control over dysregulated PKA signaling by Gα$_s$^{R201C}-expressing (Gα$_s$^{R201C+}) cells and thus allow more precise dissection of this oncogenic signaling network than would otherwise be possible. We generated two dual-stable cell lines: one A-KINACT effector cell line stably expressing A-KINACT and integrating TetO-driven EGFP-fused PKA inhibitory peptide (TetO-PKI-EGFP), and one A-KINACT control cell line stably expressing A-KINACT and integrating TetO-EGFP. Characterization of both revealed efficient responses to blue light illumination and PKA activator treatment, or Gα$_s$^{R201C}-mediated high PKA activity (Supplementary Fig. 16). The A-KINACT effector memorizes elevated PKA activity and induces the expression of PKI-EGFP to inhibit PKA activity, which should perturb the balance between PKA and ERK signaling. A-KINACT-expressing HEK293T cells showing mTagBFP2-Gα$_s$^{R201C} overexpression and A-KINACT-driven PKI-EGFP expression were identified and collected via triple-color fluorescence-activated cell sorting (FACS, Supplementary Fig. 17 and 18) to assess differences in transcription and cell proliferation versus control cells expressing A-KINACT-driven EGFP (Fig. 5a).

To obtain differentially expressed genes (DEGs), we first identified overall transcriptional events from Gα$_s$^{R201C} by comparing Gα$_s$^{R201C+}/EGFP$^+$ cells to Gα$_s$^{R201C-}/EGFP$^-$ cells, followed by contrasting with the effects of normalizing PKA activity on overall transcriptional events from Gα$_s$^{R201C} by comparing Gα$_s$^{R201C+}/PKI-EGFP$^+$ cells and Gα$_s$^{R201C-}/PKI-EGFP$^-$ cells. Analysis of differentially expressed genes (DEGs) from RNAseq data identified 1064 DEGs (Supplementary Data 3) that were upregulated in Gα$_s$^{R201C+}/PKI-EGFP$^+$ cells compared with Gα$_s$^{R201C-}/PKI-EGFP$^-$ cells, whereas 410 DEGs (Supplementary Data 4) were upregulated in Gα$_s$^{R201C+}/EGFP$^+$ cells compared with Gα$_s$^{R201C-}/EGFP$^-$ cells (Supplementary Fig. 19). We then performed transcription factor (TF) motif enrichment analysis to predict enriched PKA- or ERK-activated TF binding sites within the promoters of DEGs using the software HOMER[33]. The results from motif enrichment analysis indicated that DEGs upregulated by Gα$_s$^{R201C} contained promoter motifs that are predicted to bind PKA-activated TFs (e.g., TBP, ATF1, CREB and NFY[34–37]) or ERK-activated TFs (e.g., ETS-family (ETS:RUNX, ETV2, ETV1, ETS1-distal) and YY1[38]) (Fig. 5b). As expected, fewer PKA-activated TFs were predicted (only ATF1) in cells with the activation of PKI negative feedback. Strikingly, in the presence of the PKI negative feedback, many more ETS-family TFs, as well as YY1, Sp-family transcription factors[39,40] and PR[41], which are reported to be ERK effectors, were enriched through motif analysis (Fig. 5c). Looking into genes that were upregulated as the result of Gα$_s$^{R201C} in the presence of PKI feedback inhibition (PKI-expressing cells), we found that only 15 ATF1-associated genes (1.4% of total DEGs) could be assigned to be upregulated via PKA-activated TF, indicating that A-KINACT-mediated PKI induction robustly inhibited PKA-mediated transcription. However, 731 genes (68.7%)

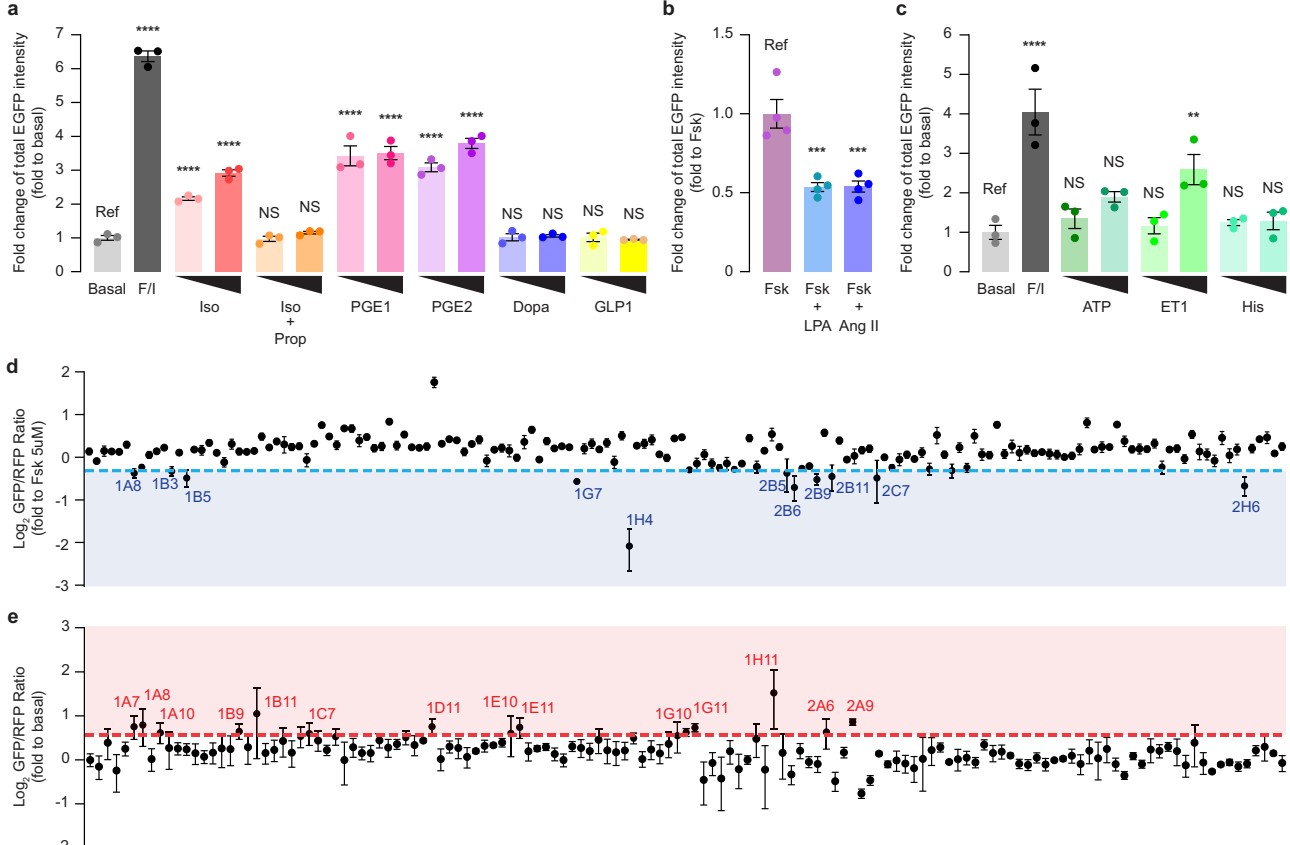

**Fig. 4 | A-KINACT for evaluating PKA responses to small molecule libraries.**
**a** Functional PKA responses to $G\alpha_s$-coupled receptor agonists: Iso (100 nM, 1 μM); Iso (100 nM, 1 μM) with propranolol (Prop) (10 μM) costimulation; PGE1 (1 μM, 10 μM); PGE2 (1 μM, 10 μM); dopamine (Dopa) (1 μM, 10 μM); glucagon-like peptide 1 (GLP1) (10 nM, 30 nM). ****$P < 1.00 \times 10^{-20}$ (F/I), ****$P = 9.76 \times 10^{-6}$ (Iso 100 nM), ****$P = 3.21 \times 10^{-11}$ (Iso, 1 μM), ****$P = 2.33 \times 10^{-15}$ (PGE1, 1 μM), ****$P = 2.33 \times 10^{-14}$ (PGE1, 10 μM), ****$P = 4.35 \times 10^{-13}$ (PGE2, 1 μM) and ****$P < 1.00 \times 10^{-20}$ (PGE2, 10 μM).
**b** Functional PKA responses to $G\alpha_i$-coupled receptor agonists: Fsk (1 μM) costimulation with LPA (500 nM) or angiotensin II (Ang-II; 10 μM). ***$P = 0.0006$ (LPA) and ***$P = 0.0007$ (Ang II). **c** Functional PKA responses to $G\alpha_q$-coupled receptor agonists: ATP (1 μM, 10 μM); endothelin 1 (ET1) (30 nM, 100 nM) and histamine (His) (1 μM, 10 μM). **$P = 0.0083$ and ****$P = 7.68 \times 10^{-6}$. **d** High-throughput library screening of 160 kinase inhibitors to discover potential PKA inhibitors. Compounds with an average value below the 0.8-fold cut-off (blue dashed line) were collected. **e** High-throughput library screening of 137 marine natural products to discover potential PKA activators. Compounds with an average value above the 1.5-fold cut-off (red dashed line) were collected. For (**a**–**e**) data from 3 independent experiments. For (**a**–**c**) statistical analysis was performed using ordinary one-way ANOVA followed by Dunnett's multiple-comparisons test. NS, not significant. Data are mean ± s.e.m. Source data are provided as a Source Data file.

were predicted to be upregulated by ERK (Fig. 5d). By contrast, in terms of genes upregulated as the result of $G\alpha_s^{R201C}$ (in A-KINACT control cells), 106 genes (25.9%) were predicted to be upregulated by PKA, and 75 genes (18.3%) were predicted to be activated by ERK (Fig. 5d). Additionally, GO enrichment analysis of the 106 PKA-upregulated genes from $G\alpha_s^{R201C}$ cells with high PKA activity (i.e., A-KINACT control cells) highlighted genes that are known to negatively regulate the ERK cascade (GO: 0070373), such as ATF3, DUSP1, DUSP4 and DUSP10 (Supplementary Fig. 19 and 20). To demonstrate the reliability of our analytical method, we verified some of the identified genes that were upregulated under high PKA activity mediated by PKA catalytic domain via quantitative PCR (Supplementary Fig. 21). These findings suggest that while $G\alpha_s^{R201C}$ upregulates both PKA- and ERK-mediated transcription, PKA activity antagonizes and suppresses the ERK-regulated transcriptome. To further validate this finding, we evaluated the signaling alterations through immunoblotting of phospho-CREB and phospho-ERK levels to examine PKA activity and ERK activation, respectively. We found that overexpressing $G\alpha_s^{R201C}$ increased both CREB and ERK phosphorylation, whereas PKI effector induction reduced phosphorylated CREB accumulation but further increased ERK phosphorylation (Supplementary Fig. 22), corroborating our findings at the transcription level.

GO enrichment analysis of the 731 ERK-activated genes identified in the presence of PKI feedback inhibition revealed many biological functions, including cell cycle modulation (Fig. 5e). Given the important role of ERK in cell proliferation, we hypothesize that normalizing PKA activity in $G\alpha_s^{R201C}$ cells through PKA-activity-driven feedback inhibition will effectively relieve inhibition of the ERK pathway and further enhance ERK-mediated cell proliferation. To test this hypothesis, we performed a cell proliferation assay by counting cells for 7 days. Indeed, PKI effector-expressing cells showed enhanced proliferation compared with A-KINACT control cells (Fig. 5f), while no difference in cell growth rate was observed between the two conditions without light induction (Supplementary Fig. 23). Thus, $G\alpha_s^{R201C}$ is capable of strongly activating the ERK pathway, which can directly contribute to increased cell proliferation. This hyperactive ERK signaling is fortuitously suppressed by activated PKA, inhibition of which leads to further boosting of the pro-proliferative ERK signaling. These findings provide critical insights into the signaling effects of an oncogenic mutation that influences a key signaling network and have important implications for therapeutic interventions.

## Discussion
Activity integrators were first conceived to address the challenges of using real-time biosensors to perform large-scale recording of

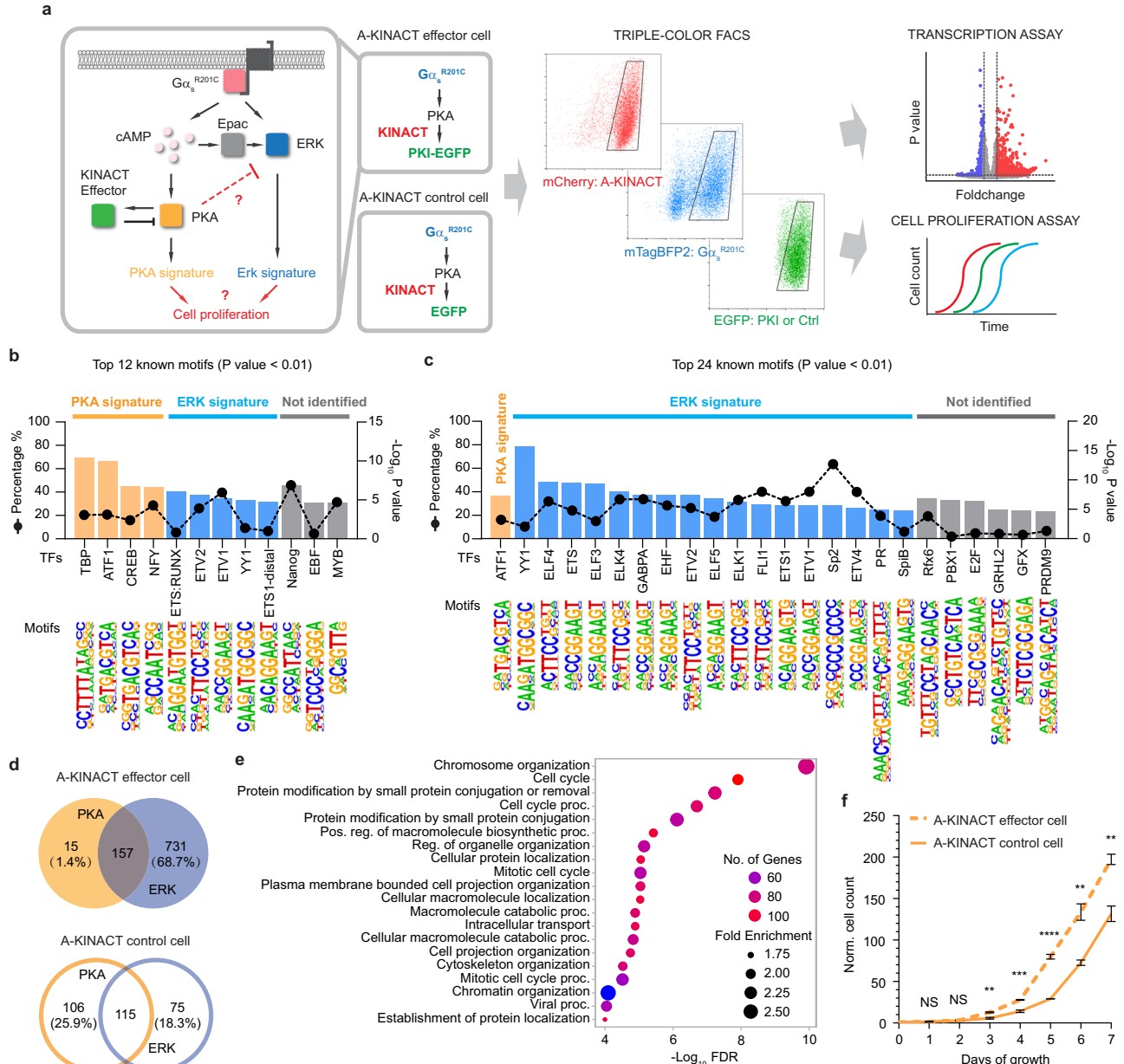

**Fig. 5 | A-KINACT effector diverts Gα$_s^{R201C}$-PKA signaling toward the ERK pathway. a** Design of A-KINACT effector for manipulating Gα$_s^{R201C}$-mediated PKA activity and scheme for identifying transcriptional and phenotypic changes. **b** Top 12 enriched known TF motifs among 410 upregulated DEGs from light-induced A-KINACT control (EGFP) cells overexpressing Gα$_s^{R201C}$. **c** Top 24 enriched known TF motifs among 1064 upregulated DEGs from light-induced A-KINACT effector (PKI-EGFP) cells overexpressing Gα$_s^{R201C}$. **d** Venn diagrams comparing the PKA signature versus ERK signature from A-KINACT effector cells (upper) and A-KINACT control

cells (lower). **e** GO enrichment analysis of 731 ERK-upregulated DEGs for biological processes. **f** Growth curves of light-reduced A-KINACT effector cells (dashed line) and A-KINACT control cells (solid line) with Gα$_s^{R201C}$ overexpression. **P = 0.005 (day 3), ***P = 0.00099 (day 4), ****P = 0.00007 (day 5), **P = 0.0042 (day 6) and **P = 0.0043 (day 7). Statistical analyses were performed using unpaired two-tailed Student's t-tests. NS, not significant. Data from 3 independent experiments. Data are mean ± s.e.m. Source data are provided as a Source Data file.

neuronal activity to identify functional neural circuits in the brain[2]. While the earliest of these activity integrators were indeed developed and implemented to record Ca$^{2+}$ (i.e., neuronal activity)[3,4] and neuromodulator dynamics[5], the utility of these tools extends beyond neuroscience, leading to their generalization to monitor processes such as protein-protein interactions between cell-surface receptors and intracellular adapters[7,42] and cell-cell contacts[43]. Here, we further demonstrate the generality of this powerful concept by reporting the development of KINACT, the first integrator for recording protein kinase activity. KINACT incorporates a dual, optical- and biochemical-switch design to enable recording of kinase activity during precisely

defined windows. We show that KINACT is capable of sensitively and specifically memorizing changes in kinase activity in response to diverse stimuli and at different subcellular locations. Although the KINACT system consists of two constructs, one of which is 5 kb, raising some potential concerns, it should be generally feasible to deliver and express the KINACT system in various model systems using viral infection, in a similar fashion as the Ca$^{2+}$ integrators[3,4]. Because its design features the same kinase-inducible switch, based on the interaction of a kinase-specific substrate and PAABD, found in real-time kinase sensors[6,11,12], KINACT is also readily adapted to probe the activities of a multitude of kinases, as we demonstrate with the Ser/Thr

kinases PKA (A-KINACT) and PKC (C-KINACT), as well as the Tyr kinase Fyn (F-KINACT). KINACT thus marks a substantial addition to the growing toolkit of activity integrators.

KINACT offers several notable advantages over real-time kinase sensors. First, KINACT accumulates activity signals over time, which are further amplified by the transcriptional readout. We believe that this feature creates new opportunities for robust and sensitive discovery of kinase activators or inhibitors. Indeed, using A-KINACT, we were able to identify a number of kinase inhibitors with potential inhibitory effects against PKA. Eight of these inhibitors are reported to function as ATP-competitive inhibitors; thus, it is not surprising that they exhibit cross-activity with other kinases. Notably, Afuresertib (Akt) and Staurosporine (multiple kinases) inhibited PKA activity even more potently than H89, hinting at the need to exercise caution when applying these inhibitors in cells. On the other hand, the fact that three non-ATP-competitive inhibitors, namely, Miransertib (Akt), CRT0066101 (PKD) and AG879 (ETK/BMX), were also able to inhibit PKA activity may indicate crosstalk between these kinases and PKA signaling. More excitingly, we discovered several potentially novel PKA activators among a panel of marine natural products. Psammaplin A is known to inhibit class I histone deacetylase (HDAC1). A broad-spectrum HDAC inhibitor, Trichostatin A (TSA), has been reported to prevent the deacetylation of PKA catalytic subunits, thereby increasing PKA activity[44]. Future work will determine whether Psammaplin A modulates PKA activity in a similar fashion. In addition, treating cells with the ETC complex 1 inhibitor Kalkitoxin led to a robust induction of PKA activity within 20 min, suggesting a distinct, rapid pathway linking respiratory chain disruption to PKA signaling. Interestingly, two Cathepsin B inhibitors, Crossbyanol B and (7Z,9Z,12Z)-octadeca-7,9,12-trien-5-ynoic acid, were both identified using A-KINACT, implying that the molecular functions of Cathepsin B, such as the proteolysis of extracellular matrix components and disruption of intercellular communication, may modulate PKA activity. Complementing its high sensitivity, another advantage of KINACT is the flexibility of readout options. In addition to fluorescent proteins, the reporter can be easily substituted with luciferase[42] or secreted alkaline phosphatase (SEAP)[5]. The latter two modalities would be more ideally suited for compound screening given the intrinsic fluorescence of many drug-like molecules.

Another major advantage of the KINACT system is that it allows *post hoc* analysis of kinase-activity-marked cell populations. Characterizing kinase activities in the complex tumor microenvironment is challenging yet critical for understanding the roles of various kinases in tumor-stroma-immune interactions, and guiding kinase-targeting therapeutics. KINACT provides a platform for addressing this challenge and for studying the spatial distribution of kinase activity within large tissues, such as tumors. Using this capability, we observed upregulated basal PKA activity in 3D versus 2D cultures. Future studies will investigate the underlying mechanisms, with the mechano-sensitivity of the PKA pathway as a possible avenue[45].

Finally, the KINACT system offers the capacity for kinase-activity-driven manipulation of targeted cell populations. Using this capability, we engineered a feedback circuit to use the recorded high PKA activity to drive the expression of a PKA inhibitor to "normalize" cellular PKA activity levels. This circuit allowed us to dissect the functional effects of the oncogenic $G\alpha_s^{R201C}$ mutant via ERK signaling, which is antagonized by PKA. PKA has been shown to inhibit the ERK pathway in multiple ways, such as blocking Ras-dependent signals to ERK by blocking Raf-1 activation[46,47], or indirectly regulating EPAC through a feedback loop controlling the production and degradation of cAMP[48,49]. We found that ATF3, DUSP1, DUSP4 and DUSP10 were upregulated by PKA, which could be involved in dephosphorylation and inactivation of the ERK family. This finding provides further insight into the interactions between PKA and ERK signaling pathways. It is worth noting that A-KINACT activates a feedback loop that exerts an inhibitory effect only in cells with high PKA activity. PKA activity levels are thus normalized to a relatively uniform level between cells, minimizing functional differences, which is difficult to achieve by simply overexpressing PKI. A previous study has shown that $G\alpha_s^{R201C}$ signals through a PKA-Salt Inducible Kinase (SIK) pathway to initiate pancreatic ductal adenocarcinoma[29]. Although PKA can promote tumorigenesis[50], we show that neutralizing PKA activity can also increase cell proliferation by relieving inhibition of ERK signaling also induced by hyperactive $G\alpha_s^{R201C}$. Similarly, a recent study showed blocking PKA signaling in mutant $G\alpha_s$ cells resulted in tumor aggression[51]. Thus, signaling activities driven by oncogenic mutations can be complex, and using KINACTs to advance our understanding of both the complete repertoire of signaling activities driven by oncogenic mutations and the complex cross-regulation between the major players would be critical for developing effective therapeutic strategies.

## Methods

### Generating KINACT

A-KINACT components 1 and 2 were cloned based on blue-light-inducible TEV protease (BLITz) constructs (ref. [5]) by Gibson assembly. Briefly, to generate A-KINACT constructs 1–5, CIBN in BLITz component 1 was replaced with a PKA substrate sequence (LRRATLVD). For constructs 1–3, hLOV1 domain-tethered TEVseq-G, TEVseq-P or TEVseq-M were introduced to replace the iLID-TEVseq-G in BLITz component 1. For construct 5, eLOV-TEVseq-M amplified by PCR from FLARE (ref. [4]) was used to replace iLID-TEVseq-G. For construct 6, TEVp-N was removed from construct 1. To generate A-KINACT construct 7, CRY2PHR in BLITz component 2 was replaced with an FHA1 domain. A-KINACT construct 8 was generated based on construct 7 by replacing TEVp-C with full-length TEVp. Mitochondrial outer membrane-targeted A-KINACT component 1 was generated by replacing the PDGFβ transmembrane helix with the N-terminal 30 amino acids from DAKAP1 (MAIQLRSLFPLALPGMALLGWWWFFSRKK). C-KINACT and F-KINACT were generated based on A-KINACT by replacing the PKA substrate with PKC or Fyn substrates, respectively, as well as replacing FHA1 with SH2 domain (for F-KINACT). Non-phosphorylatable negative-control substrate mutants were generated by PCR. Polycistronic A-KINACT, C-KINACT and F-KINACT constructs were assembled following the order component 1-P2A-mCherry-IRES-component 2. Lentiviral backbones containing Bleomycin (FUGW, Addgene #14883) or Hygromycin (w117-1, Addgene #17454) resistance genes were used to generate lentiviral constructs of A-KINACT and A-KINACT (T/A). The TetO-H2B-EGFP and TetO-PKI-EGFP reporter constructs were cloned by replacing EGFP in pTRE-EGFP (Addgene # 89871) with H2B-EGFP and PKI-EGFP, respectively, via Gibson assembly. A lentiviral backbone containing a Puromycin (w118-1, Addgene #17452) resistance gene was used to generate lentiviral reporter constructs. The DNA or protein sequences of all constructs are provided in the Supplement.

### Other plasmids

AKAR4[13], ExRai-AKAR2 (ref. [9]), GR-AKARev (ref. [16]), Booster-PKA (ref. [17]), and AKAR3ev (ref. [24]) were described previously. mTagBFP2-GNAS(R201C) was generated in a pcDNA3 backbone by Gibson assembly. All constructs were verified by Sanger sequencing (Genewiz).

### Cell culture and transfection

HEK293T, HeLa and MCF-7 cells were cultured in Dulbecco's modified Eagle medium (DMEM; Gibco) and supplemented with 10% (v/v) fetal bovine serum (FBS, Sigma) and 1% (v/v) penicillin-streptomycin (Pen-Strep, Sigma-Aldrich). All cells were maintained at 37 °C in a humidified atmosphere of 5% $CO_2$. For imaging experiments, cells were plated onto sterile 35 mm glass-bottomed dishes (Cellvis) and grown to 50%

confluence. Transient transfection was performed using PolyJet (SignaGen) for HEK cells or Lipofectamine 2000 (Invitrogen) for HeLa and MCF-7 cells, after which cells were cultured for an additional 24 h before blue light illumination or imaging.

## Stable cell lines

To package the lentivirus containing TetO-H2B-EGFP, TetO-EGFP, TetO-PKI-EGFP or A-KINACT, each pLenti-construct was mixed with psPAX2 and pMD2.G-VSV-g (3:2:1) and co-transfected into HEK293T cells. The medium containing packaged lentivirus was collected after 48 h and then concentrated using Lenti-X™ Concentrator (Clontech). To generate TetO-H2B-EGFP, TetO-EGFP and TetO-PKI-EGFP stable cell lines, concentrated solution containing lentiviral particles was added to HEK293T, HeLa or MCF-7 cells. After 48 h, cells were diluted and plated in 10 cm cell culture dishes and then treated with Puromycin (1 μg ml⁻¹) to select transduced cells. Single colonies were picked after a week and characterized by transfecting tTA to induce TetO-promoted gene expression. To generate A-KINACT or A-KINACT (T/A) plus TetO-gene dual-stable cell lines, concentrated lentiviral particles containing A-KINACT or A-KINACT (T/A) were added to established TetO-H2B-EGFP, TetO-EGFP and TetO-PKI-EGFP stable HEK293T or MCF-7 cells. A-KINACT + TetO-H2B-EGFP dual-stable cells were selected with Puromycin (2 μg ml⁻¹) and Zeocin (Invitrogen, 300 μg ml⁻¹). A-KINACT (T/A) + TetO-H2B-EGFP, A-KINACT + TetO-EGFP and A-KINACT + TetO-PKI-EGFP dual-stable cells were selected with Puromycin (1 μg ml⁻¹) and Hygromycin (Invitrogen, 150 μg ml⁻¹). Single colonies were picked after a week and characterized by Fsk/IBMX and blue light treatment to induce reporter gene expression.

## Blue-light illumination, snapshot imaging and image analysis

Blue-light illumination was performed using a 465-nm-wavelength blue LED (SunLED, XLFBB12W) array controlled by a high-accuracy electronic timer (GraLab, model 451). Glass-bottom 35 mm dishes or 96-well plates were raised 1.5 cm above the LED source to avoid potential undesirable heating caused by direct contact with the LED. After 24 h expression, cells were stained with Hoechst 33342 for 20 min at 37 °C and washed once with Hanks's balanced salt solution (HBSS, Gibco) prior to imaging. Cells were imaged in HBSS on a Zeiss AxioObserver Z1 microscope equipped with a 20 × /0.8 NA objective and a Hamamatsu Flash 4.0 sCMOS camera, controlled using a modified version of the open-source MATscope suite[52] implemented in MATLAB 2017a (Mathworks Inc.). Hoechst 33342 was imaged using a 420DF20 excitation filter, a 455DRLP dichroic mirror, and a 470DF40 emission filter. EGFP was imaged using a 495DF10 excitation filter, a 515DRLP dichroic mirror, and a 535DF25 emission filter. mCherry was imaged using a 568DF55 excitation filter, a 600DRLP dichroic mirror, and a 653DF95 emission filter.

Images were processed using Cell Profiler and segmented using DAPI channel images. To calculate normalized total EGFP fluorescence for each condition, the integrated EGFP intensity of all H2B-EGFP⁺ cells in a well was summed and then divided by the number of mCherry⁺ (i.e., KINACT-expressing) cells. To calculate the G/R ratio of individual cells, the mean EGFP intensity of each H2B-EGFP⁺ cell was divided by its own mean mCherry intensity.

## Immunoblotting

Cells were washed with ice-cold PBS and then lysed in RIPA lysis buffer containing protease inhibitor cocktail, PMSF (1 mM), Na3VO4 (1 mM), NaF (1 mM), and calyculin A (25 nM). Total cell lysates were incubated on ice for 30 min and then centrifuged at 4 °C for 20 min. Total protein was separated via 4-15% SDS-PAGE and transferred to PVDF membranes. The membranes were blocked with TBS containing 0.1% Tween-20 and 5% bovine serum albumin and then incubated with primary antibodies overnight at 4 °C. After incubation with the

appropriate horseradish peroxidase (HRP)-conjugated secondary antibodies, the membranes were developed using horseradish peroxidase-based chemiluminescent substrate (34579 and 34076, Thermoscientific). The intensity of the bands was quantified with ImageJ software. The following primary antibodies were used for immunoblotting: Myc-Tag (#2278), CREB (#9197), Phospho-CREB (Ser133) (#9198), p44/p42 MAPK (Erk1/2, #4695) and Phospho-p44/42 MAPK (Erk1/2, Thr202/Tyr204) (#9106) antibodies from Cell Signaling Technology. The HRP-labeled goat anti-rabbit (PI31460) or anti-mouse (PI31430) secondary antibodies were purchased from Pierce.

## Immunofluorescence

Cells expressing A-KINACT or Mito-A-KINACT were washed 3 times with PBS and fixed with 4% paraformaldehyde (15710-S, Electron Microscopy Sciences) in PBS for 10 min at room temperature. For Mito-A-KINACT localization assay, cells were washed 3 times with PBS and permeabilized with PBS containing 0.1% Triton X-100 for 10 min at room temperature. For A-KINACT localization assay, cells were only washed 3 times with PBS without permeabilization in order to avoid high intracellular background caused by plasma membrane marker. Following 1 h incubation in blocking buffer (PBS containing 5% BSA) at room temperature, cells were incubated for 1 h at 4 °C with Myc-Tag (#2278, Cell Signaling Technology) primary antibody diluted in blocking buffer (1:500). Following three 5 min washes in PBS, cells were incubated for 1 h at room temperature in the dark with anti-rabbit secondary antibody conjugated Alexa Fluor™ 488 (A11006, Invitrogen) diluted in PBS (1:1000). Following three 5 min washes with PBS, cells were labeled with MitoTracker™ Red FM (M22425, Invitrogen) or MemGlow™ 640 Fluorogenic Membrane Probe (MG04, Cytoskeleton, Inc.) for 10 min before confocal imaging on Leica SP8.

## Spheroid culture and 3D imaging

Matrigel (Growth Factor Reduced Matrigel, BD Biosciences) was plated onto 35 mm glass-bottom dishes and allowed to solidify for 30 min at 37 °C. Dual-stable HEK293T or MCF-7 cells were trypsinized, resuspended in media supplemented with 2% Matrigel and then plated on top of Matrigel. After 5 days in culture, spheroids were induced by blue light for 20 min and cultured for another 24 h before imaging. Z-stack imaging was performed on a Leica SP8 confocal equipped with a 20 × / 0.75 NA objective over a total range of 60 μm (3 μm step per Z-position). The 405 nm, 488 nm and 561 nm laser bands were chosen for excitation of DAPI, H2B-EGFP and mCherry, respectively. DAPI, H2B-EGFP and mCherry emission were collected using 420–460 nm band-pass, 500–540 nm band-pass and 600-650 nm band-pass filters, respectively. 3D images were reconstructed using Leica software LAS X. An attenuation correction was applied to confocal image stacks to compensate for the decrease of recorded fluorescence signal intensities with depth. An ImageJ plug-in implementing correction algorithm is available at https://imagejdocu.list.lu/plugin/stacks/attenuation_correction/start. Attenuation correction was applied independently to the DAPI, RFP and GFP channels.

## Fluorescence microplate screening

Black-walled, glass-bottom 96-well assay plates (Costar) were coated with 0.1 mg ml⁻¹ of poly-D-lysine in boric acid buffer (pH 8.5) for 30 min at 37 °C and then seeded with 5 × 10⁴ A-KINACT + TetO-H2B-EGFP dual-stable HEK293T cells per well for 24 h culturing. For PKA inhibitor screening, cells were treated with kinase inhibitor library (Cayman chemical, #10505-0577971, 10 μM each drug) and blue light for 20 min (10 s on/50 s off), followed by 24 h expression. For PKA activator screening, cells were treated with marine natural products (1 μg ml⁻¹ per drug) and blue light for 20 min (10 s on/50 s off), followed by 24 h expression.

Fluorescence intensity was read on a Spark 20 M fluorescence plate reader using SparkControl Magellan 1.2.20 software (TECAN).

Cells were washed once and then placed in HBSS to acquire an RFP-GFP sequential read. mCherry was read at 555 nm excitation/640 nm emission and EGFP was read at 485 nm excitation/535 nm emission. Five regions were read per well. Raw green and red fluorescence values from each region were corrected by subtracting the background fluorescence intensity of a nontransfected well and then averaged. G/R ratio in each well was calculated by dividing the average EGFP intensity by the average mCherry intensity. Three replicates were performed.

### Time-lapse epifluorescence imaging

HEK293T cells expressing AKAR4, ExRai-AKAR2, GR-AKARev, Booster-PKA or AKAR3ev were washed twice with HBSS and subsequently imaged in HBSS in the dark at RT. Forskolin, IBMX, H89, Iso, PGE1, PGE2, LPA, Angiotensin II (Ang II), ATP, endothelin 1 (ET-1), Histamine (His), 11 screened kinase inhibitors and 4 selected marine natural products were added as indicated. Cells were imaged on a Zeiss AxioObserver Z1 microscope (Carl Zeiss) equipped with a $40 \times /1.3$ NA objective and a Photometrics Evolve 512 EMCCD (Photometrics) controlled by METAFLUOR 7.7 software (Molecular Devices). For ExRai-AKAR2, dual GFP excitation-ratio imaging was performed using 480DF30 and 405DF40 excitation filters, a 505DRLP dichroic mirror and a 535DF45 emission filter. For GR-AKARev, dual green/red emission ratio imaging was performed using a 480DF30 excitation filter, a 505DRLP dichroic mirror and two emission filters (535DF45 for green fluorescent protein and ET605DF52 for red fluorescent protein). For Booster-PKA, dual orange/red emission ratio imaging was performed using a 555DF25 excitation filter, a 568DRLP dichroic mirror and two emission filters (605DF52 for orange fluorescent protein and 650DF100 for red fluorescent protein). For AKAR3ev, dual cyan/yellow emission ratio imaging was performed using a 420DF20 excitation filter, a 450DRLP dichroic mirror and two emission filters (475DF40 for cyan fluorescent protein and 535DF25 for yellow fluorescent protein). Filter sets were alternated by a Lambda 10-2 filter-changer (Sutter Instruments). Exposure times ranged between 50 and 500 ms, with EM gain 10, and images were acquired every 30 s. For the time-series analysis, all values were normalized to the time point prior to drug addition by dividing each value by the basal ratio value prior to drug addition.

### Triple-color cell sorting by flow cytometry

A-KINACT-PKI-EGFP induced cells and A-KINACT-EGFP control cells were collected and washed once with $1 \times$ DPBS, resuspended in HBSS buffer containing 1% FBS and strained (FALCON) to prevent clogging. Both A-KINACT-PKI-EGFP and A-KINACT-EGFP cells were sorted for mTagBFP2 at Ex. 405 nm (FL1 450/50 emission filter), EGFP at Ex. 488 nm (FL2 525/50 emission filter) and mCherry at Ex. 560 nm (FL3 600/60 emission filter) on a SONY SH800S Cell Sorter running Cell Sorter Software (Ver 2.1.6). Debris, dead cells, and cell aggregates were gated out before fluorescence interrogation by monitoring the forward and side scatter (Supplementary Fig. 13). Approximately 500,000 single cells were collected for each sample. Sorting was repeated three times.

### RNA-sequencing and data analysis

Total RNA from one aliquot of sorted cells was extracted using an RNA extraction kit (Zymol), and 20 μl of 50 ng μl$^{-1}$ sample was sent to Novogene following the company's instruction. Samples that passed quality control were proceeded to paired-end mRNA sequencing. Raw data were analyzed using the web-based platform Galaxy (https://usegalaxy.org) and open-source R software (version 4.1.0, https://www.r-project.org). Briefly, high-quality raw reads were mapped to the reference human genome hg38 using HISAT2 (version 2.2.1), and aligned reads were counted using htseq-count (version 0.9.1). The R package DESeq2 (version 1.34.0) was used to conduct differentially

expressed gene (DEG) analysis. Genes with fold change (>2) and $P$-adj <0.05 were assigned as significantly altered genes.

Within each set of DEGs, upregulated genes were subjected to motif enrichment analysis using Homer (version 4.11). Briefly, the findMotifs.pl function in HOMER was used to analyze the promoters of DEGs and look for motifs that are enriched in these promoters (ref. 33). The known motifs with $P$-value < 0.01 from top 25 motifs and corresponding binding sites for a certain TF were retained. The predicted TFs were classified to PKA or ERK pathways according to literatures (refs. 34–39,41.). To find instances of specific motifs, annotatePeaks.pl function was run with the parameter of size -300, 50. Gene ontology (GO) enrichment analysis was performed using the online tool ShinyGO 0.77 (http://bioinformatics.sdstate.edu/go).

### qPCR

HEK293T cells with or without overexpression of PKA catalytic domain were collected. Total RNA was then isolated using TRIzol reagent, and 1 μg of RNA was reverse-transcribed into cDNA using the PrimeScript RT master mix (TAKARA). qPCR analysis was carried out using iTaq universal SYBR green supermix (Bio-Rad). Each sample measurement was performed in triplicate. The following sense and antisense primers were used. ATF3: sense, CGC TGG AAT CAG TCA CTG TCA G; antisense, CTT GTT TCG GCA CTT TGC AGC TG; DUSP1: sense, CAA CCA CAA GGC AGA CAT CAG C; antisense, GTA AGC AAG GCA GAT GGT GGC T; DUSP4: sense, TAC TCG GCG GTC ATC GTC TAC G; antisense, CGG AGG AAA ACC TCT CAT AGC C; DUSP10: sense, CAG CCA CTT CAC ATA GTC CTC G; antisense, TGG AGG GAG TTG TCA CAG AGG T. Relative mRNA levels were quantified using the comparative Ct method after normalization to GAPDH.

### Cell proliferation assay

One aliquot of the sorted cells from FACS was seeded in 12-well plates at 10,000 cells per well (Day 0). Cell numbers were quantified using a Countess II cell counter (Life Technologies) each day for 7 days.

### Statistics and reproducibility

All experiments were independently repeated as noted in the figure legends. Statistical analyses were performed using GraphPad Prism 8. For Gaussian data, pairwise comparison of two parametric data sets was performed using Student's $t$-test. For comparing three or more sets of data, ordinary one-way ANOVA followed by the indicated multiple comparisons test was done. Statistical significance was defined as $P < 0.05$ with a 95% confidence interval.

### Reporting summary

Further information on research design is available in the Nature Portfolio Reporting Summary linked to this article.

## Data availability

The RNA sequencing data generated in this study have been deposited in the Gene Expression Omnibus (GEO) database under accession code GSE272741. The remaining data are available within the Article, Supplementary Information or Source Data file. Source data are provided with this paper.

## Code availability

Custom MATLAB code used for image acquisition and analysis is available at https:// github.com/jinzhanglab-ucsd/MatScopeSuite (https://doi.org/10.5281/zenodo.5908008).

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

## Acknowledgements

The authors are grateful to all members of the Zhang lab for various technical assistance and thoughtful discussions. We especially thank Dr Eric. C. Greenwald (now at Pfizer) for assembling the LED blue light array and Professor Jing Yang (UCSD) for providing MCF-7 cells. We also wish to thank Dr. Jason Z. Zhang (now at U of Washington) for generating mTagBFP2-GNAS (R201C) and Dr. Evgenia Glukhov for preparing the marine natural products library. Work in J.Z.'s laboratory is supported by the National Institutes of Health (NIH) (R35 CA197622, R01 DK073368 and R01 CA262815 to J.Z.). Work in Y.W.'s laboratory is supported by the National Institutes of Health (R01 EB029122, R35 GM140929, R01 HD107206 and R01 CA262815 to Y.W.). The MNP library from W.H.G.'s laboratory is supported by the National Institutions of Health (R01 GM107550 to W.H.G) and the Chancellor of UC San Diego. This work is also supported by the UCSD Microscopy Core (P30 NS047101).

## Author contributions

W.L., A.P., and J.Z. conceived the project. W.L. and A.P. developed KINACT. W.L. generated stable cell lines and performed light illumination and live-cell epifluorescence imaging, spheroid culturing and 3D confocal imaging, high-throughput library screening, western blotting, RNAseq, qPCR, cell proliferation assays, and data processing. Y.Z. analyzed the RNA sequencing data. L.L. performed the cell sorting experiment. H-B.K. provided BLITz constructs. W.H.G. provided the marine natural product library. Y.W. provided the kinase inhibitor library. Y.W., S.M., and J.Z. supervised the project and coordinated experiments. W.L., S.M., and J.Z. wrote the manuscript with input from all authors.

## Competing interests

The authors declare no competing interests.
