## [Peer Review File · Nature Communications]

Reviewers' Comments:

Reviewer #1:

Remarks to the Author:

Lin et al present a new system for converting short-lived elevations in PKA activity into permanent fluorescent marks: A-KINACT. The development strategy that they have employed is very rigorous and systematic, and the basic design of the reporter is well thought-through. After demonstrating that the reporter works for PKA (Fig. 1), they show that it can be easily adapted for monitoring other kinases (Fig. 2). They go on to apply A-KINACT to image organoids (Fig. 3), to perform high-throughput screening (Fig 4), and finally to investigate interplay between PKA and ERK downstream of oncogenic GalphaS mutation (Fig. 5). It could be argued that components of Fig. 4/5 could have been performed with existing technologies, but nevertheless (i) this new reporter is novel and does offer advantages compared to ratiometric FRET-based reporters of PKA phosphorylation (ii) the organoid imaging is novel and there are many biological systems where this type of imaging could be performed, e.g. the brain (iii) the high-throughput screening has thrown up interesting results, e.g. the effect of ET1 & identification of murine compounds that inhibit PKA (iv) the paper supports the intriguing finding that PKA may serve an 'anti-oncogenic' function downstream of pathological ERK activation. Overall, I found the manuscript to be well-written with detailed and comprehensive methods. I have only a few minor comments/questions:

For Supp fig. 1b - should the third column be labelled '+Fsk/IBMX / -light'?

For figure 1 panels 1 & f – it would help to label WT and T/A, which was not immediately apparent without digging into the legend.

In figure. 1g, and also in Fig. 4a, iso triggers only a modest elevation in PKA phosphorylation even at high concentrations. Does this imply that only a small fraction of PKA is activated with full b-AR activation? I was surprised that PGE1/2 elicited a greater response. Is this finding consistent with earlier reports using FRET-based PKA/cAMP reporters?

Figure 1j. There is a more pronounced change evoked by light only in this location. Does this imply that basal PKA phosphorylation at the mitochondrial outer membrane is higher than at the plasma membrane?

Figure 3. For organoid imaging (Fig. 3), have the authors attempted equivalent imaging with AKAR4? (I wonder how it would compare to reveal whether elevated PKA activity is sustained or transient).

Regarding the organoid imaging, is there any pattern to GFP+ cells using A-KINACT? It looks like there may be more EGFP+ near to the outside of the organoid and in the upper layers.

Reviewer #2:

Remarks to the Author:

The authors developed an “integrator” of kinase activity, i.e. a reporter containing an optical and a biochemical switch that enables recording of kinase activity. They focused on PKA, but also showed that their approach is applicable to other kinases, including serine/threonine and tyrosine kinases. Finally, they leveraged this system to manipulate PKA activity in a feedback manner to probe intersecting signaling pathways (PKA and ERK) activated by a mutant form of Gas G protein subunit, GasR201C.

The manuscript is detailed, clearly written and the engineering part, together with all controls and validating experiments is thorough.

On the other hand the experiments in support of the two most attractive applications that could be enabled by this system, proposed here, are less compelling and the related data not so convincing. More specifically:

First, the authors propose that this system could enable recording of kinase activity in 3D organoids and complex tissues. However the proof of principle experiment shown is very underwhelming – a 3D culture of 293T cells, which is very far removed from an organoid culture.

This brings the related, but critical, concern regarding the feasibility of delivery and expression of these very complex and large constructs in relevant cell types, that would be needed for more realistic and useful applications of this system, such as adult stem cells or pluripotent stem cells. Many engineering approaches work beautifully in 293T cells but are not transferrable in more interesting or relevant cell culture systems.

Second, the experiments described in Fig. 5 lack robustness to support the main claim that “These findings suggest that GasR201C upregulates both PKA- and ERK-mediated transcription and yet PKA activity antagonizes ERK signaling and suppresses the ERK-regulated transcriptome.”

Specific comments- Major:

1. The analyses presented in Fig. 5 are difficult to evaluate. It is unclear to me why the authors compared GasR201C+/PKI-EGFP+ cells to GasR201C-/PKI-EGFP- cells and not to GasR201C+/PKI-EGFP- or GasR201C+/EGFP+. The latter would allow to validate that

PKI indeed efficiently works to inhibit PKA and would appear to better separate PKA from ERK signaling activation by GasR201C.

In addition, the assessment of PKA and ERK pathway activation by gene expression changes is indirect and inconclusive and does not allow firm conclusions. Biochemical assessment of the activation status of these pathways (pERK, pMEK, PKA westerns etc) is lacking.

2. The claims about “organoids” is vastly exaggerated. 3D cultures of 293T cells are not by any stretch an “organoid”.

Additional comments:

3. The analytical methodology used to assign TF motifs to gene promoters and to classify TFs in PKA/ERK pathways is not described and justified.

In addition, statements such as “PKA-activated TFs were reduced” and “many more ETS-family TFs...were upregulated” are factually incorrect, as the TFs themselves were not reduced or upregulated.

4. The authors suggest that A-KINACT can be targeted to plasma membrane or mitochondria to examine local PKA activity. However, these subcellular localizations are not verified by immunofluorescence.

5. In addition to A-KINACT, they constructed C-KINACT and F-KINACT. While they perform confirmatory experiments to establish that A- and C-KINACT are dependent on PKA and PKC activity, respectively, Fyn-dependency of F-KINACT was not validated.

6. The experiments presented in Figure 3 and Supplemental Fig.6b are interpreted to reveal that HEK293 cells in 3D culture show heterogeneous PKA activity (Figure3). However, the flow cytometry analysis shown in Supplemental Fig.6b does not show reflect this heterogeneity – if anything the histograms shown suggest that EGFP expression in 3D is more homogeneous than in 2D.

7. I am unsure of the use of the term “permanent” in many instances, since reporter EGFP expression is transient, not permanent.

Minor comments

- Specify what the control cells in fig 5f are.
- “which coverts calcium transients”: the rod converts is misspelled
- Explain what the FHA1 domain of AKAR is.
- TEVp, TEV, TEVseq, TEVs: please use consistent names

- SFig 2: show the plasmid used to transfect 293T cells for this quantification by WB (is the MYC tag before or after the P2A in component 1?)
- How many vector copies does the stable 293T line with integrated TetO-H2B-EGFP reporter contain?
- The data shown in Supplementary Figure 8 can be better explained.
- In Figure 4, the authors mention that dopamine, GLP1 and histamine receptors are absent in HEK293T. References should be included.
- In line 3, page 11, Supplemental Fig. 15 should be Supplemental Fig. 14.

Reviewer #3:

Remarks to the Author:

Kinase signaling pathways are key regulators of cellular processes, both under normal physiological conditions and during tumorigenesis and cancer development. Fluorescent biosensors that report on kinase activity in real time have been developed, but the recording of kinase signaling activity for post-hoc analysis is relatively less explored. Lin et al. developed a light-gated system that converts kinase activity to permanent fluorescence using a transcriptional reporter. The design, optimization, and validation of the system are robust and thorough, and the application of the system in high-throughput screening for kinase activators and inhibitors is both innovative and impactful. However, the authors described additional applications of the KINACT system, including in visualizing the spatial heterogeneity of kinase activity and in exploring the complex mechanisms of Gas signaling, which need additional support and clarification.

Specific comments:

1. The use of “WT” as a designation is misleading, since A-KINACT is an engineered construct and therefore there is no “wildtype” naturally expressed in cells. The authors should consider renaming the groups to “A-KINACT” and “A-KINACT(T/A)” or something similar to make it clear that both are engineered cell lines.
2. The mito-targeted A-KINACT reported lower change in GFP fluorescence upon stimulation compared to the plasma membrane-targeted A-KINACT, which the authors interpret as an organelle-dependent difference in PKA activity change. Is the assumption that A-KINACT has the same sensitivity on both membranes? Can the use of a different targeting tag change the orientation of the TEV protease and/or PKAsub/FHA1 in such a way that sensitivity is reduced?
3. Fig. 3a: the later Z slices appear dimmer than the earlier ones, which indicates that

the samples may be too thick. Can cells deep inside the culture activate KINACT properly, or is KINACT activation restricted to cells on the periphery where blue light is able to penetrate?

4. Supplementary Fig. 6: although the GFP signal of the 3D culture is higher than the 2D culture, indicating higher PKA activity, the RFP signal of the 3D culture is also much higher than the 2D culture, presumably because of the thickness of the culture in the z direction. It would be a more fair comparison if confocal images or possibly z-stack of both cultures were shown.

5. What is the rationale of using AKAR3ev for validating the PKA activators while GR-AKARev and Booster-PK were used to validate the PKA inhibitors?

6. The results of the PKI-EGFP study is interesting and the analysis of PKA vs ERK signature TF motifs is impressive. However, in Supplementary Fig. 13 it appears that most of the cells in Gate F were positive for GFP in both groups, so the KINACT readout really had no impact on the outcome of this experiment. Additionally, using the KINACT system to induce PKI expression seems like a convoluted method for inducing PKA inhibition. Wouldn't it be more robust to simply overexpress PKI or using a pharmaceutical inhibitor? The authors should consider clarifying the experimental design and/or expanding the study to link the Gas signaling back to PKA activity (for example using the high throughput protocol described earlier in the text to validate some of the TF hits shown in Fig. 5, seeing if overexpression of those TFs can directly upregulate PKA activity).

RESPONSES TO REVIEWER COMMENTS

Reviewer #1 (Remarks to the Author):

Lin et al present a new system for converting short-lived elevations in PKA activity into permanent fluorescent marks: A-KINACT. The development strategy that they have employed is very rigorous and systematic, and the basic design of the reporter is well thought-through. After demonstrating that the reporter works for PKA (Fig. 1), they show that it can be easily adapted for monitoring other kinases (Fig. 2). They go on to apply A-KINACT to image organoids (Fig. 3), to perform high-throughput screening (Fig 4), and finally to investigate interplay between PKA and ERK downstream of oncogenic GalphaS mutation (Fig. 5). It could be argued that components of Fig. 4/5 could have been performed with existing technologies, but nevertheless (i) this new reporter is novel and does offer advantages compared to ratiometric FRET-based reporters of PKA phosphorylation (ii) the organoid imaging is novel and there are many biological systems where this type of imaging could be performed, e.g. the brain (iii) the high-throughput screening has thrown up interesting results, e.g. the effect of ET1 & identification of murine compounds that inhibit PKA (iv) the paper supports the intriguing finding that PKA may serve an 'anti-oncogenic' function downstream of pathological ERK activation. Overall, I found the manuscript to be well-written with detailed and comprehensive methods. I have only a few minor comments/questions:

We thank the Reviewer for the thoughtful comments.

For Supp fig. 1b - should the third column be labelled '+Fsk/IBMX / -light'?

We apologize for the confusion. In Supp Figure 1b, the two columns on the left are "single component" under dark or light conditions without drug treatment. For the two columns on the right, we compare no drug/no light negative condition to drug/light positive condition in cells co-expressing "two components" to identify constructs that give the lowest background and the best fold change. We have modified figure legend in Supplementary Figure 1 to make it clearer.

For figure 1 panels 1 & f - it would help to label WT and T/A, which was not immediately apparent without digging into the legend.

We thank the Reviewer for this suggestion and have now added labels accordingly.

In figure. 1g, and also in Fig. 4a, iso triggers only a modest elevation in PKA phosphorylation even at high concentrations. Does this imply that only a small fraction of PKA is activated with full b-AR activation? I was surprised that PGE1/2 elicited a greater response. Is this finding consistent with earlier reports using FRET-based PKA/cAMP reporters?

We thank the Reviewer for this comment. We have validated these conditions using AKAR4, which showed iso induced responses were more transient than PGE1/2. Although the maximal response to iso was similar to PGE1/2 (shown by AKAR), A-KINACT reports the integrated response where iso-induced response appeared to be modest, due to its transient nature, and smaller than that induced by PGE1/2. The validation has been added to Supplementary Figure 12.

Figure 1j. There is a more pronounced change evoked by light only in this location. Does this imply that basal PKA phosphorylation at the mitochondrial outer membrane is higher than at the plasma membrane?

We thank the Reviewer for this comment. It has been shown that there is a higher basal PKA activity at mitochondrial outer membrane^{1,2}. We also measured the basal and stimulated PKA activities close to plasma membrane and on outer mitochondrial membrane using TM-ExRaiAKAR2 and DAKAP-ExRaiAKAR2, respectively (the same targeting motifs in A-KINACT and Mito-A-KINACT). TM-ExRaiAKAR2 reported a lower starting ratio ($R_0=0.35$) than DAKAP-ExRaiAKAR2 ($R_0=1.09$), while the maximal change of response to Fsk/IBMX close to PM was much higher ($\Delta R_{\max}/R_0=1.11$) than on OMM ($\Delta R_{\max}/R_0=0.31$) (Reviewer Fig. 1).

Figure 3. For organoid imaging (Fig. 3), have the authors attempted equivalent imaging with AKAR4? (I wonder how it would compare to reveal whether elevated PKA activity is sustained or transient).

We thank the Reviewer for this suggestion. We generated spheroids with HEK293T cells stably expressing ExRai-AKAR2, a single FP-based AKAR, and imaged the middle layer of each spheroid. We observed heterogenous PKA activities, which were sustained within 20 min (Reviewer Fig. 2). The data is in general agreement with the A-KINACT result. We did not perform long-term imaging so it is difficult to know exactly how long the elevated activity lasts; but A-KINACT should be able to “mark” any of the active cells during the light gating, irrespective of how transient or sustained the activity is.

Regarding the organoid imaging, is there any pattern to GFP+ cells using A-KINACT? It looks like there may be more EGFP+ near to the outside of the organoid and in the upper layers.

We thank the Reviewer for this comment. The reduced EGFP intensity of the inside of the organoid may be due to light scattering and absorption as the imaging depth increases. We applied an ImageJ Attenuation Correction plugin to correct the fluorescence intensity of each channel (DAPI, GFP and RFP) and included the representative intensity-corrected images in Supplementary Figure 10. After correction, the distribution of H2B-EGFP+ cells appeared to be random in each layer, with no obvious difference between upper and lower layers.

Reviewer #2 (Remarks to the Author):

The authors developed an “integrator” of kinase activity, i.e. a reporter containing an optical and a biochemical switch that enables recording of kinase activity. They focused on PKA, but also showed that their approach is applicable to other kinases, including serine/threonine and tyrosine kinases. Finally, they leveraged this system to manipulate PKA activity in a feedback manner to probe intersecting signaling pathways (PKA and ERK) activated by a mutant form of Gas G protein subunit, GasR201C.

The manuscript is detailed, clearly written and the engineering part, together with all controls and validating experiments is thorough.

On the other hand the experiments in support of the two most attractive applications that could be enabled by this system, proposed here, are less compelling and the related data not so convincing. More specifically:

We thank the Reviewer for the thoughtful comments. We have performed additional experiments and modified the text to address these comments.

First, the authors propose that this system could enable recording of kinase activity in 3D organoids and complex tissues. However the proof of principle experiment shown is very underwhelming – a 3D culture of 293T cells, which is very far removed from an organoid culture.

This brings the related, but critical, concern regarding the feasibility of delivery and expression of these very complex and large constructs in relevant cell types, that would be needed for more realistic and useful applications of this system, such as adult stem cells or pluripotent stem cells. Many engineering approaches work beautifully in 293T cells but are not transferrable in more interesting or relevant cell culture systems.

We thank the Reviewer for this comment. As our lab currently does not have access to adult stem cells or pluripotent stem cells, we have focused on demonstrating this system in cancer cell spheroids, which is also aligned with the cancer focus of later figures. We applied the A-KINACT integrator in the MCF-7 breast cancer cell line. We generated MCF-7+TetO-H2B-EGFP+A-KINACT and MCF-7+TetO-H2B-EGFP+A-KINACT(T/A) dual-stable cell lines and cultured them into spheroids. 3D imaging was performed to map the distribution of H2B-EGFP+ cells within spheroids, which indicate cells with high basal PKA activities. We added these 3D imaging data in Supplementary Figure 11. Our data shows that the KINACT system could be used in cancer spheroids, which represents a useful model system. Although the KINACT system consists of two constructs, one of which is 5 kb, raising some potential concerns, it should be generally feasible to deliver and express the KINACT system in various model systems using viral infection. In fact, similar integrator systems for calcium and neuromodulators, such as FLARE³ and iTango⁴, have been used in brain tissues through adeno-associated viruses. We have added a discussion about this point (Page 13). We also have a plan to write a protocol paper to detail step-by-step procedures for delivering and using the KINACT system to encourage its testing in other model systems.

Second, the experiments described in Fig. 5 lack robustness to support the main claim that “These findings suggest that GasR201C upregulates both PKA- and ERK-mediated

transcription and yet PKA activity antagonizes ERK signaling and suppresses the ERK-regulated transcriptome.”

We thank the Reviewer for this comment. We have performed new experiments and clarified the analysis strategies. These changes have been incorporated into the text (Page 12) and Supplementary Figure 22.

Specific comments- Major:

1. The analyses presented in Fig. 5 are difficult to evaluate. It is unclear to me why the authors compared GasR201C+/PKI-EGFP+ cells to GasR201C-/PKI-EGFP- cells and not to GasR201C+/PKI-EGFP- or GasR201C+/EGFP+. The latter would allow to validate that PKI indeed efficiently works to inhibit PKA and would appear to better separate PKA from ERK signaling activation by GasR201C.

We apologize for the confusion. Our goal was to identify transcriptional events mediated by $G\alpha_s^{R201C}$ and to provide some insights into the transcriptional impact by the complex PKA-ERK cross-regulated network. If we were to only compare $G\alpha_s^{R201C+}/PKI-EGFP+$ cells to $G\alpha_s^{R201C+}/EGFP+$ cells, both PKA-mediated events and ERK-mediated events (via PKA-regulated ERK) may be manifested – they may antagonize each other resulting in unclear differences. Indeed, this comparison yielded few DEGs. We decided to first identify overall transcriptional events from $G\alpha_s^{R201C}$ by comparing $G\alpha_s^{R201C+}/EGFP+$ cells to $G\alpha_s^{R201C-}/EGFP-$ cells; then contrast with the effects of normalizing PKA activity on overall transcriptional events from $G\alpha_s^{R201C}$ (difference between $G\alpha_s^{R201C+}/PKI-EGFP+$ cells and $G\alpha_s^{R201C-}/PKI-EGFP-$ cells) to deduce the effects mediated by PKA as well as ERK. Although a bit complex, our analysis did yield new findings suggesting that $G\alpha_s^{R201C}$ upregulates both PKA- and ERK-mediated transcription and yet PKA activity antagonizes ERK signaling and suppresses the ERK-regulated transcriptome.

One of our findings is that a few negative regulators of ERK are upregulated by PKA, one of which, DUSP1, was also identified by a recent publication to be upregulated by PKA⁵. We also validated some of the identified genes (ATF3, DUSP1, DUSP4, and DUSP10) by testing their expression changes under overexpression of PKA catalytic domain by qPCR. We have now modified the text (Page 11-12) to further clarify our strategy and included the validation data in Supplementary Figure 21.

In addition, the assessment of PKA and ERK pathway activation by gene expression changes is indirect and inconclusive and does not allow firm conclusions. Biochemical assessment of the activation status of these pathways (pERK, pMEK, PKA westerns etc) is lacking.

We thank the Reviewer for the comments. As the reviewer suggested, we performed western blotting assay to quantify phosphorylation of CREB and ERK. We observed that pCREB and pERK were both increased when $G\alpha_s^{R201C}$ was overexpressed in EGFP control cells. In PKI effector cells overexpressing $G\alpha_s^{R201C}$, pCREB was reduced but pERK was further increased when PKI effector was induced, supporting our conclusion that $G\alpha_s^{R201C}$ upregulates both PKA and ERK activities and yet PKA activity antagonizes and suppresses the ERK signaling. We added the data in Supplementary Figure 22.

2. The claims about "organoids" is vastly exaggerated. 3D cultures of 293T cells are not by any stretch an "organoid".

We thank the Reviewer for the comments. We have changed the wording to "spheroids".

Additional comments:

3. The analytical methodology used to assign TF motifs to gene promoters and to classify TFs in PKA/ERK pathways is not described and justified.

We thank the Reviewer for the comments. We first used the findMotifs.pl function in HOMER to analyze the promoters of DEGs and look for motifs that are enriched in these promoters⁶. The predicted TFs were classified to PKA or ERK pathways according to literatures⁷⁻¹³. We then ran annotatePeaks.pl function to find instances of specific motifs. We have added relevant information to the main text (Page 11) and Methods section.

In addition, statements such as “PKA-activated TFs were reduced” and “many more ETS-family TFs....were upregulated” are factually incorrect, as the TFs themselves were not reduced or upregulated.

We have changed the wording to “fewer PKA-activated TFs were predicted” and “many more ETS-family TFs, ...were enriched through motif analysis”, respectively, in the main text (Page 11).

4. The authors suggest that A-KINACT can be targeted to plasma membrane or mitochondria to examine local PKA activity. However, these subcellular localizations are not verified by immunofluorescence.

We thank the Reviewer for the suggestion and have added immunofluorescence data in Supplementary Figure 6.

5. In addition to A-KINACT, they constructed C-KINACT and F-KINACT. While they perform confirmatory experiments to establish that A- and C-KINACT are dependent on PKA and PKC activity, respectively, Fyn-dependency of F-KINACT was not validated.

We thank the Reviewer for the comment. We have validated kinase-dependency of C-KINACT and F-KINACT by using PKC inhibitor and Fyn inhibitor, respectively. The data was added to Figure 2.

6. The experiments presented in Figure 3 and Supplemental Fig.6b are interpreted to reveal that HEK293 cells in 3D culture show heterogeneous PKA activity (Figure3). However, the flow cytometry analysis shown in Supplemental Fig.6b does not show reflect this heterogeneity – if anything the histograms shown suggest that EGFP expression in 3D is more homogeneous than in 2D.

We thank the Reviewer for this comment. We have changed the wording to highlight the increased population of cells showing high PKA activity in 3D instead of increased heterogeneity.

7. I am unsure of the use of the term “permanent” in many instances, since reporter EGFP expression is transient, not permanent.

Permanent is a relative term meant to contrast with transient, real-time readouts. In the short time period (i.e. 24 hr) between recording PKA activity and post hoc measurement, the induced reporter expression was sufficiently persistent. We also added quotes to indicate the so-called “permanent”.

Minor comments

We thank the Reviewer very much for the following suggestions and corrections. All of them have now been edited or explained accordingly.

- *Specify what the control cells in fig 5f are.*

We have uniformized the name of control cells as “A-KINACT control cells” and specified the stable cell line in the main text.

- *“which converts calcium transients”: the rod converts is misspelled.*

The correction has been made.

- *Explain what the FHA1 domain of AKAR is.*

The explanation of FHA1 domain has been added (Page 3).

- *TEVp, TEV, TEVseq, TEVs: please use consistent names*

All TEV-related names have been uniformized in the main text and figures.

- *SFig 2: show the plasmid used to transfect 293T cells for this quantification by WB (is the MYC tag before or after the P2A in component 1?)*

The Myc-epitope tags were fused to N-termini of component 1 and 2, which has been indicated in figure legend. The amino acid sequences are shown at the end of supplemental information. We have added schematic domain structures beside the WB.

- *How many vector copies does the stable 293T line with integrated TetO-H2B-EGFP reporter contain?*

Although we found some protocols for determining transgene copy numbers by qPCR^{14,15}, we were not able to obtain accurate copy numbers without a control cell line with known transgene copy numbers. To provide more characterization of the integrated TetO-H2B-EGFP reporter systems, we evaluated the expression ability of the inducible reporters by directly transfecting and expressing the tTA transcription factor. The different cell lines used in this manuscript, including HEK293T, HeLa and MCF-7, can all successfully integrate TetO-H2B-EGFP reporter into the genome and show good expression. The data was added to Supplementary Fig. 3.

- *The data shown in Supplementary Figure 8 can be better explained.*

Explanations have been added to the main text (Page 9).

- *In Figure 4, the authors mention that dopamine, GLP1 and histamine receptors are absent in HEK293T. References should be included.*

We referred to RNAseq data for HEK293T on Gene Expression Omnibus (GEO) database¹⁶ which shows absence or low expression of these GPCRs. We added the link of GEO database and primary reference in the main text.

- *In line 3, page 11, Supplemental Fig. 15 should be Supplemental Fig. 14.*

Both previous Supplementary Fig. 14 and 15 (now changed to Supplementary Fig. 19 and 20) support the statement. The text has been updated to cite both figures.

Reviewer #3 (Remarks to the Author):

Kinase signaling pathways are key regulators of cellular processes, both under normal physiological conditions and during tumorigenesis and cancer development. Fluorescent biosensors that report on kinase activity in real time have been developed, but the recording of kinase signaling activity for post-hoc analysis is relatively less explored. Lin et al. developed a light-gated system that converts kinase activity to permanent fluorescence using a transcriptional reporter. The design, optimization, and validation of the system are robust and thorough, and the application of the system in high-throughput screening for kinase activators and inhibitors is both innovative and impactful. However, the authors described additional applications of the KINACT system, including in visualizing the spatial heterogeneity of kinase activity and in exploring the complex mechanisms of Gas signaling, which need additional support and clarification.

Specific comments:

1. The use of "WT" as a designation is misleading, since A-KINACT is an engineered construct and therefore there is no "wildtype" naturally expressed in cells. The authors should consider renaming the groups to "A-KINACT" and "A-KINACT(T/A)" or something similar to make it clear that both are engineered cell lines.

We thank the Reviewer for this comment and have now edited relevant wording.

2. The mito-targeted A-KINACT reported lower change in GFP fluorescence upon stimulation compared to the plasma membrane-targeted A-KINACT, which the authors interpret as an organelle-dependent difference in PKA activity change. Is the assumption that A-KINACT has the same sensitivity on both membranes? Can the use of a different targeting tag change the orientation of the TEV protease and/or PKAsub/FHA1 in such a way that sensitivity is reduced?

We thank the Reviewer for this concern. A flexible linker was added between targeting tags and PKAsub, which should allow some flexibility for the PKAsub to sense local PKA activity. The TEVp-N was also extended away from organelle membranes by another long linker. The component 2 containing FHA1 was diffusive in cytosol, which should not be affected by the membrane environment. Therefore, we expect that A-KINACT to have similar sensitivity on both membranes. The lower response observed on outer mitochondrial membrane is likely due to a higher basal PKA activity before stimulation^{1,2}. We also validated the high basal activity on mitochondrial outer membrane using the subcellularly targeted real-time sensor Mito-ExRaiAKAR2 compared with the plasma membrane-targeted TM-ExRaiAKAR2. (Reviewer Fig. 1)

3. Fig. 3a: the later Z slices appear dimmer than the earlier ones, which indicates that the samples may be too thick. Can cells deep inside the culture activate KINACT properly, or is KINACT activation restricted to cells on the periphery where blue light is able to penetrate?

We thank the Reviewer for this comment. For the blue light illumination, we used 465 nm LEDs (SunLED, XLFBB12W) with strong luminous intensity (5990 millicandela), which should be sufficient to illuminate the entire spheroid. All three channels (DAPI, GFP and

RFP) showed reduced fluorescence intensities in the later Z slices, likely due to light scattering and absorption as the imaging depth increases. We applied an ImageJ Attenuation Correction plugin to correct the fluorescence intensity of each channel and found clear GFP signal in the corrected images, confirming that the blue light was able to penetrate into the spheroid to turn on KINACT. We included the representative intensity-corrected images in Supplementary Figure 10.

4. Supplementary Fig. 6: although the GFP signal of the 3D culture is higher than the 2D culture, indicating higher PKA activity, the RFP signal of the 3D culture is also much higher than the 2D culture, presumably because of the thickness of the culture in the z direction. It would be a more fair comparison if confocal images or possibly z-stack of both cultures were shown.

We thank the Reviewer for this suggestion. We have now used confocal imaging for both 2D and 3D samples and included the data in Supplementary Figure 8.

5. What is the rationale of using AKAR3ev for validating the PKA activators while GR-AKARev and Booster-PK were used to validate the PKA inhibitors?

We thank the Reviewer for this question. We chose a set of conventional FRET-based AKARs to validate these potential PKA activators and inhibitors. As the PKA activators identified by A-KINACT-based screening show relatively weak effects, we used AKAR3ev which is the most sensitive PKA activity reporter at present. For inhibitor validation, most of inhibitors show blue or blue-green fluorescence so that we chose GR-AKARev or Booster-PKA to avoid interference by the compound fluorescence. This also shows that it is difficult to complete a full set of high-throughput compound screening with any real-time fluorescent biosensor, while the integrator with post hoc measurement can effectively avoid the interference from the compound fluorescence. We have now added this explanation to the text (page 10).

6. The results of the PKI-EGFP study is interesting and the analysis of PKA vs ERK signature TF motifs is impressive. However, in Supplementary Fig. 13 it appears that most of the cells in Gate F were positive for GFP in both groups, so the KINACT readout really had no impact on the outcome of this experiment.

We thank the Reviewer for these comments.

We apologize for the confusion. What is observed here – that most cells in Gate F are positive for EGFP in both groups – is what we expected. Both EGFP and PKI-EGFP expression are induced by high PKA activity via A-KINACT, so most cells with $G\alpha_s^{R201C}$ overexpression (in Gate F) will indeed show increased EGFP or PKI-EGFP expression (e.g., Supplementary Fig. 16). We used Gate I-1 to collect EGFP⁺ or PKI-EGFP⁺ cells with fluorescence intensity greater than 10^4 (over 80%) in order to exclude cells with weak PKA responses (e.g. Supplementary Fig. 18). EGFP⁺ or PKI-EGFP⁺ cells were then used for RNAseq experiments. We have now added negative control cells ($G\alpha_s^{R201C}$ -/PKI-EGFP- and $G\alpha_s^{R201C}$ -/EGFP-) to better illustrate the gating strategy in Supplementary Fig. 17 and 18.

Additionally, using the KINACT system to induce PKI expression seems like a convoluted method for inducing PKA inhibition. Wouldn't it be more robust to simply overexpress PKI or using a pharmaceutical inhibitor?

Simply overexpressing PKI may lead to excessive inhibition effect. This system is set up to directly “interact” with existing PKA activity within the cell and to self-modulate, such that only cells with hyperactive PKA should be inhibited by this system and, importantly, the degree of inhibition is controlled by the actual PKA activity level. We expect that such an approach would lead to less toxicity than would otherwise arise from PKA inhibition in normal cells with lower, but nevertheless biologically important, levels of PKA activity.

The authors should consider clarifying the experimental design and/or expanding the study to link the Gas signaling back to PKA activity (for example using the high throughput protocol described earlier in the text to validate some of the TF hits shown in Fig. 5, seeing if overexpression of those TFs can directly upregulate PKA activity).

Sorry for the confusion. We have now included additional discussions of the experimental design. Our goal was to identify transcriptional events mediated by $G\alpha_s^{R201C}$ and to provide some insights into the transcriptional impact by the complex PKA-ERK cross-regulated network. We showed that $G\alpha_s^{R201C}$ upregulates both PKA- and ERK-mediated transcription and yet PKA activity antagonizes ERK signaling and suppresses the ERK-regulated transcriptome.

One of our findings is that a few negative regulators of ERK are upregulated by PKA, some of which, for example DUSP1, were also identified by a recent publication reporting PKA transcriptional signatures⁵. As encouraged by the Reviewer, we have also expanded the study to validate some of our hits (ATF3, DUSP1, DUSP4 and DUSP10) by testing their expression changes under overexpression of PKA catalytic domain by qPCR. We included the validation data in Supplementary Figure 21.

References

- 1 Allen, M. D. & Zhang, J. Subcellular dynamics of protein kinase A activity visualized by FRET-based reporters. *Biochem. Biophys. Res. Commun.* **348**, 716-721, doi:10.1016/j.bbrc.2006.07.136 (2006).
- 2 Langenmayer, I. *et al.* Engraftment of patients with lymphoid malignancies transplanted with autologous bone marrow, peripheral blood stem cells or both. *Bone Marrow Transplant.* **15**, 241-246 (1995).
- 3 Wang, W. *et al.* A light- and calcium-gated transcription factor for imaging and manipulating activated neurons. *Nat. Biotechnol.* **35**, 864-871, doi:10.1038/nbt.3909 (2017).
- 4 Lee, D. *et al.* Temporally precise labeling and control of neuromodulatory circuits in the mammalian brain. *Nat. Methods* **14**, 495-503, doi:10.1038/nmeth.4234 (2017).
- 5 Burghi, V. *et al.* Galphas is dispensable for beta-arrestin coupling but dictates GRK selectivity and is predominant for gene expression regulation by beta2-adrenergic receptor. *J. Biol. Chem.* **299**, 105293, doi:10.1016/j.jbc.2023.105293 (2023).
- 6 Heinz, S. *et al.* Simple combinations of lineage-determining transcription factors prime cis-regulatory elements required for macrophage and B cell identities. *Mol. Cell* **38**, 576-589, doi:10.1016/j.molcel.2010.05.004 (2010).
- 7 Cluck, M. W., Murphy, L. O., Olson, J., Knezetic, J. A. & Adrian, T. E. Amylin gene expression mediated by cAMP/PKA and transcription factors HNF-1 and NFY. *Mol. Cell. Endocrinol.* **210**, 63-75, doi:10.1016/j.mce.2003.08.005 (2003).
- 8 Kinane, T. B., Shang, C., Finder, J. D. & Ercolani, L. cAMP regulates G-protein alpha i-2 subunit gene transcription in polarized LLC-PK1 cells by induction of a CCAAT box nuclear binding factor. *J. Biol. Chem.* **268**, 24669-24676 (1993).
- 9 Baler, R., Covington, S. & Klein, D. C. The rat arylalkylamine N-acetyltransferase gene promoter. cAMP activation via a cAMP-responsive element-CCAAT complex. *J. Biol. Chem.* **272**, 6979-6985, doi:10.1074/jbc.272.11.6979 (1997).
- 10 Cote, F. *et al.* Involvement of NF-Y and Sp1 in basal and cAMP-stimulated transcriptional activation of the tryptophan hydroxylase (TPH) gene in the pineal gland. *J. Neurochem.* **81**, 673-685, doi:10.1046/j.1471-4159.2002.00890.x (2002).
- 11 Zheng, H., Chu, J., Zeng, Y., Loh, H. H. & Law, P. Y. Yin Yang 1 phosphorylation contributes to the differential effects of mu-opioid receptor agonists on microRNA-190 expression. *J. Biol. Chem.* **285**, 21994-22002, doi:10.1074/jbc.M110.112607 (2010).
- 12 Zassadowski, F., Rochette-Egly, C., Chomienne, C. & Cassinat, B. Regulation of the transcriptional activity of nuclear receptors by the MEK/ERK1/2 pathway. *Cell. Signal.* **24**, 2369-2377, doi:10.1016/j.cellsig.2012.08.003 (2012).
- 13 Terrados, G. *et al.* Genome-wide localization and expression profiling establish Sp2 as a sequence-specific transcription factor regulating vitally important genes. *Nucleic Acids Res.* **40**, 7844-7857, doi:10.1093/nar/gks544 (2012).
- 14 Grandjean, M. *et al.* High-level transgene expression by homologous recombination-mediated gene transfer. *Nucleic Acids Res.* **39**, e104, doi:10.1093/nar/gkr436 (2011).
- 15 Shin, S., Kim, S. H., Lee, J. S. & Lee, G. M. Streamlined Human Cell-Based Recombinase-Mediated Cassette Exchange Platform Enables Multigene Expression for the Production of Therapeutic Proteins. *ACS Synth Biol* **10**, 1715-1727, doi:10.1021/acssynbio.1c00113 (2021).
- 16 Edgar, R., Domrachev, M. & Lash, A. E. Gene Expression Omnibus: NCBI gene expression and hybridization array data repository. *Nucleic Acids Res.* **30**, 207-210, doi:10.1093/nar/30.1.207 (2002).

Reviewers' Comments:

Reviewer #1:

Remarks to the Author:

The reviewers have very rigorously addressed my comments.

Reviewer #2:

Remarks to the Author:

The authors made several revisions in response to my and the other reviewers' critiques. They appropriately replaced the word "organoids" with "cancer spheroids". In response to my concerns regarding the experiments presented in Fig. 5, the authors explained and justified satisfactorily their analytical approach. They also added Westerns to complement transcriptional data for demonstration of activation of PKA and ERK pathways (although pCREB and pERK by western blots are quite underwhelming). Overall the authors were very responsive and made a big effort to address all criticisms and the manuscript is improved as a consequence.

Reviewer #3:

Remarks to the Author:

The authors have adequately addressed my concerns and I now support publication of the paper as is. I congratulate the authors on an excellent study and a useful new tool.

REVIEWERS' COMMENTS

Reviewer #1 (Remarks to the Author):

The authors have very rigorously addressed my comments.

We appreciate the Reviewer's kind comments.

Reviewer #2 (Remarks to the Author):

The authors made several revisions in response to my and the other reviewers' critiques. They appropriately replaced the word "organoids" with "cancer spheroids". In response to my concerns regarding the experiments presented in Fig. 5, the authors explained and justified satisfactorily their analytical approach. They also added Westerns to complement transcriptional data for demonstration of activation of PKA and ERK pathways (although pCREB and pERK by western blots are quite underwhelming). Overall the authors were very responsive and made a big effort to address all criticisms and the manuscript is improved as a consequence.

We are pleased the Reviewer found our revisions to strengthen the findings reported in this paper and thank the Reviewer for their time and effort.

Reviewer #3 (Remarks to the Author):

The authors have adequately addressed my concerns and I now support publication of the paper as is. I congratulate the authors on an excellent study and a useful new tool.

We are pleased the Reviewer found our revisions to be satisfactory and recommend the paper for publication. We thank the Reviewer for their time and effort.